# Towards a Unified and Verified Understanding of Group-Operation Networks

**Wilson Wu**[1]   **Louis Jaburi**[*2]   **Jacob Drori**[*2]   **Jason Gross**[2]
[1] University of Colorado Boulder    [2] Independent
`wiwu2390@colorado.edu`
`{louis.yodj,jacobcd52,jasongross9}@gmail.com`

## Abstract

A recent line of work in mechanistic interpretability has focused on reverse-engineering the computation performed by neural networks trained on the binary operation of finite groups. We investigate the internals of one-hidden-layer neural networks trained on this task, revealing previously unidentified structure and producing a more complete description of such models in a step towards unifying the explanations of previous works (Chughtai et al., 2023; Stander et al., 2024). Notably, these models approximate *equivariance* in each input argument. We verify that our explanation applies to a large fraction of networks trained on this task by translating it into a *compact proof of model performance*, a quantitative evaluation of the extent to which we *faithfully* and *concisely* explain model internals. In the main text, we focus on the symmetric group $S_5$. For models trained on this group, our explanation yields a guarantee of model accuracy that runs 3x faster than brute force and gives a $\geq 95\%$ accuracy bound for 45% of the models we trained. We were unable to obtain nontrivial non-vacuous accuracy bounds using only explanations from previous works.

## 1 Introduction

Modern neural network models, despite their widespread deployment and success, remain largely inscrutable in their inner workings, limiting their use in safety-critical settings. The emerging field of *mechanistic interpretability* seeks to address this issue by reverse engineering the behavior of trained neural networks. One major criticism of this field is the lack of rigorous *evaluations* of interpretability results; indeed, many works rely on human intuition to determine the quality of an interpretation (Miller, 2019; Casper, 2023; Räuker et al., 2023). This insufficiency of evaluations has proved detrimental to interpretability research: recent work finds many commonly used model interpretations to be imprecise or incomplete (Miller et al., 2024; Friedman et al., 2024).

A simplified research program has focused on toy algorithmic settings, which are made more tractable by the presence of complete mathematical descriptions of the task and dataset (Nanda et al., 2023a;b; Zhong et al., 2024). However, even in these settings, the lack of rigorous evaluations for interpretations is consequential, leading different researchers to come up with divergent explanations for the same empirical phenomena: recently, Chughtai et al. (2023) claimed that models trained on finite groups implement a *group composition via representations* algorithm, while subsequent work (Stander et al., 2024) studies the same model and task and instead argues that the model implements a *coset concentration* algorithm.

In this work, we take on the challenge of reconciling their interpretations. We investigate the same setting and find internal model structure that was overlooked by both previous works: the irreducible representations noticed by Chughtai et al. (2023) act by permutation on a discrete set of vectors learned by the model. Based on our observations, we propose a model explanation that unifies those found by both previous works. In particular we find that the model approximates a function that preserves the group symmetry in each of its input arguments, i.e. a bi-equivariant function.

---

[*]These authors contributed equally. See Author Contributions.

Following Gross et al. (2024), we then evaluate our interpretation and that of previous work by converting them into compact proofs of lower bounds on model accuracy.

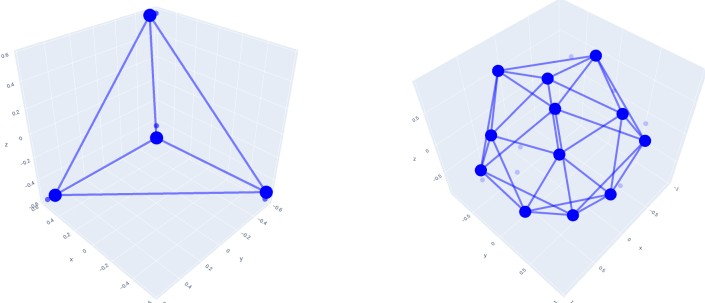

Figure 1: Examples of $\rho$-sets extracted directly from the weights of models trained on the symmetric group $S_4$ (**left**, a tetrahedron) and the alternating group $A_5$ (**right**, an icosahedron). Both lie in $\mathbb{R}^3$. The vectors of the $\rho$-sets are depicted as points—the connecting edges are merely for illustration. See Section 3 for the definition of $\rho$-sets and Section 4 for how they are in by models to compute the group operation. See Figure 10 and Figure 11 for compact proof bound results for $S_4$ and $A_5$, respectively. The standard irrep of $S_5$, the focus of the main text, is four-dimensional and hence more difficult to visualize.

Our philosophy is that any rigorous mechanistic knowledge of a model's inner workings should yield a guarantee on the model's performance. In more detail, given a mechanistic explanation, the real model will somewhat differ from it due to noise or imperfections in our analysis. To rigorously validate the explanation, we thus need to bound the effect of this deviation on model behavior. This validation is done using a program that takes in a model as input and guarantees that the model obeys some property; we focus on properties of the form "the model will give the correct answer for at least some $X\%$ of the input space", i.e. lower bounds on accuracy. The simplest such guarantee is to brute force try every possible input to the model, which does not require a mechanistic explanation and is a perfectly tight bound. However, we believe that nontrivial mechanistic understanding of a model should yield more efficient programs, e.g. by exploiting symmetries in the proposed explanation.

If the program guarantees a property such as model accuracy, then we can uniformly turn an execution trace of the program into an formal proof of this property. This is an (automatically generated) proof in the standard mathematical sense: a proof that a mathematical object (the model parameters) satisfies the desired property. Proofs corresponding to more efficient execution traces are shorter, i.e. more *compact*.[1] The efficiency of the program, and closeness of the accuracy bound to the true accuracy, can be taken as metrics of the quality of our explanation: More complete explanations should yield either tighter performance bounds or a more compact proof, providing a quantitative measure of explanation completeness. We find that our interpretation is indeed an improvement over previous ones because it yields more compact proofs of tighter bounds.

Our contributions are as follows:

- We provide a mechanistic explanation of models trained on the group composition task (Section 4).

- We *verify* this explanation by translating it into guarantees on model accuracy (Section 5). For a substantial fraction of models, these guarantees are near the true accuracy, providing strong positive evidence for our explanation in these cases (Section 5.2).

- We clarify previous mechanistic interpretability results on this same task and argue that they do not fully explain model behavior (Section 6). We show that our more complete interpretation is a step towards *unifying* the findings of previous works (Section 6.3)

---

[1]The length of the proof is linear in the running time of the program.

## 2 RELATED WORK

**Groups, mechanistic interpretability, and grokking**   Our work can be seen as a direct follow-up to Chughtai et al. (2023) and Stander et al. (2024), which both perform mechanistic interpretability on one-hidden-layer neural networks trained on the binary operation of finite groups. These papers in turn build on work that studies models trained on modular arithmetic (Nanda et al., 2023a; Zhong et al., 2024), i.e. the binary operation of the cyclic group. Models trained on group composition exhibit the *grokking* phenomenon (Power et al., 2022), in which a model trained on an algorithmic task generalizes to the test set many epochs after attaining perfect accuracy on the training set. Morwani et al. (2024) study the group composition task from the viewpoint of inductive biases, showing that, for one-hidden-layer models with quadratic activations, the max-margin solution must match the observations of Chughtai et al. (2023).

**Evaluation of explanations**   Several techniques to evaluate interpretations of models have been suggested, such as causal interventions (Wang et al., 2023) and causal scrubbing (Chan et al., 2022). We discuss merits and limitations of causal interventions in our setting, which were first explored in Stander et al. (2024). More recently, Gross et al. (2024) use mechanistic interpretability to obtain compact formal proofs of model properties for the max-of-four task. Yip et al. (2024) study the modular arithmetic setting, finding that the ReLU nonlinearities can be thought of as performing numerical integration, and use this insight to compute bounds on model error in linear time.

**Equivariance**   We find that neural networks trained on group composition learn to be equivariant in both input arguments, i.e. *bi-equivariance*, despite this condition not being enforced in the architecture. Learned equivariance has been noticed and measured in other settings (Lenc & Vedaldi, 2019; Olah et al., 2020; Gruver et al., 2023). This is distinct from the area of equivariant networks, in which equivariance is enforced by model architecture (Bronstein et al., 2021).

## 3 PRELIMINARIES

### 3.1 MATHEMATICAL BACKGROUND: GROUPS, ACTIONS, REPRESENTATIONS, $\rho$-SETS

This paper uses ideas from finite group theory and representation theory. We provide a rapid and informal introduction to the most important definitions and refer the reader to Section 3 and Appendices D, E, F of Stander et al. (2024) and/or relevant textbooks (Dummit & Foote, 2004; Fulton & Harris, 1991) for more details.

**Groups and permutations**   A group $G$ is a set with an associative binary operation $\star$ and an identity element $e$ such that every element has an inverse. We write $S_n$ for the group of permutations on $n$ elements. Maps[2] $G \to S_n$ are called *permutation representations* and are equivalent to actions of $G$ on sets of size $n$. Recall that each permutation $\sigma \in S_n$ can be represented as a *permutation matrix* in $\mathbb{R}^{n \times n}$ that applies the permutation $\sigma$ to the basis vectors of $\mathbb{R}^n$. Thus, any permutation representation of $G$ is a *linear representation*, i.e. a mapping from $G$ to the group of invertible $n \times n$ matrices $\mathrm{GL}(n, \mathbb{R})$. The group operation translates to matrix multiplication.

**Irreps**   Linear representations that cannot be decomposed into a direct sum of representations of strictly smaller dimension are called *irreducible representations* or *irreps* for short. A representation $\rho$ is an irrep if and only if there is no nontrivial subspace that is closed under $\rho(g)$ for all $g \in G$. Every finite group $G$ has a finite set of irreps (up to isomorphism), which we denote by $\mathrm{Irrep}(G)$.[3] Any linear representation can be decomposed *uniquely* into irreps.

**$\rho$-sets**   By the preceding discussion, given a permutation representation $\tilde{\rho} \colon G \to S_n$, we can consider its decomposition into irreps. Let $\rho \in \mathrm{Irrep}(G)$ be one irrep present in this decomposition, acting on some subspace $W \subseteq \mathbb{R}^n$. Since, by definition, $\tilde{\rho}$ acts on standard basis vectors $\boldsymbol{e}_1, \ldots, \boldsymbol{e}_n$ by permutation, the constituent irrep $\rho$ acts on the projection of the basis vectors onto $W$ by the same

---

[2]By "maps" we mean group homomorphisms.
[3]In this paper, we consider irreps over $\mathbb{R}$. In particular, all irreps of $S_n$ are real. For a discussion of preliminary results for groups with complex irreps, see Appendix K.2.

permutation. We refer to any subset of $W$ that $\rho$ acts on by permutation as a $\rho$-*set*; in particular, projections of the basis vectors onto $W$ fit this criterion.[4]

**Example:** $S_5$  Our primary example throughout the main text is the group of permutations $S_5$. The identity map $S_5 \rightarrow S_5$ is a permutation representation. As a linear representation, it is not irreducible, as it fixes the all ones vector $\mathbf{1} \in \mathbb{R}^5$. Projecting out this vector results in what is called the standard four-dimensional irrep of $S_5$; call it $\rho$. This irrep acts by permutation on the projections of the standard basis vectors, so these five vectors in $\mathbb{R}^4$ form a $\rho$-set. In this example, the $\rho$-set consists of five evenly spaced vectors on the surface of the sphere in $\mathbb{R}^4$.

## 3.2  TASK DESCRIPTION AND MODEL ARCHITECTURE

Our task and architecture are identical to that of previous works (Chughtai et al., 2023; Stander et al., 2024). Fix a finite group $G$. We train a model on the supervised task $\star : G \times G \rightarrow G$; i.e., given $x, y \in G$ the task is to predict the product $x \star y \in G$.

We train a one-hidden-layer two-input neural network on this task. The input to the model is two elements $x, y \in G$, embedded as vectors $\boldsymbol{E}_l(x), \boldsymbol{E}_r(y) \in \mathbb{R}^m$, which we refer to as the left and right embeddings, respectively. These are multiplied by the left and right linearities $\boldsymbol{W}_l, \boldsymbol{W}_r \in \mathbb{R}^{m \times m}$, summed, applied with an elementwise ReLU nonlinearity, and finally multiplied by the unembedding matrix $\boldsymbol{U} \in \mathbb{R}^{|G| \times m}$ and summed with the bias $\boldsymbol{w}_b$.

We can simplify the model's description by noting that the left embedding $\boldsymbol{E}_l$ and left linearity $\boldsymbol{W}_l$ only occur as a product $\boldsymbol{W}_l \boldsymbol{E}_l$, and likewise for the right embedding and linearity. Also, the product between the unembedding $\boldsymbol{U}$ and the embedding vectors can be decomposed into a sum over the hidden dimensionality $[m]$. Hence, letting $\boldsymbol{w}_l^i(x), \boldsymbol{w}_r^i(y), \boldsymbol{w}_u^i(z)$ denote the $i$th entries of $\boldsymbol{W}_l \boldsymbol{E}_l(x), \boldsymbol{W}_r \boldsymbol{E}_r(y), \boldsymbol{U}(z)$ respectively, the forward pass is

$$f_{\boldsymbol{\theta}}(z \mid x, y) = \boldsymbol{w}_b(z) + \sum_{i=1}^m \boldsymbol{w}_u^i(z) \, \text{ReLU}[\boldsymbol{w}_l^i(x) + \boldsymbol{w}_r^i(y)], \tag{1}$$

paramaterized by $\boldsymbol{\theta} = (\boldsymbol{w}_b, (\boldsymbol{w}_l^i, \boldsymbol{w}_r^i, \boldsymbol{w}_u^i)_{i=1}^m)$. Each vector $\boldsymbol{w}_l^i, \boldsymbol{w}_r^i, \boldsymbol{w}_u^i$ for $i \in [m]$ can be thought of as a function $G \rightarrow \mathbb{R}$, and we refer to them as the left, right, and unembedding neurons, respectively. The refer to $i \in [m]$ as the neuron index.

## 4  GROUP COMPOSITION BY $\rho$-SETS

### 4.1  THE $\rho$-SET CIRCUIT

Our central finding is that trained models implement circuits of the form

$$f_{\rho, \mathcal{B}}(z \mid x, y) = - \sum_{\boldsymbol{b}, \boldsymbol{b}' \in \mathcal{B}} \boldsymbol{b}^\top \rho(x^{-1} \star z \star y^{-1}) \boldsymbol{b}' \, \text{ReLU}[\boldsymbol{a}^\top (\boldsymbol{b} - \boldsymbol{b}')] \tag{2}$$

where $\mathcal{B} \subseteq \mathbb{R}^d$ is a $\rho$-set and $\boldsymbol{a} \in \mathbb{R}^d$. This $f_{\rho, \mathcal{B}}(z \mid x, y)$ depends only on $x^{-1} \star z \star y^{-1}$; we call such functions *bi-equivariant*.[5] Furthermore, we show in Lemma G.5 that for certain irreps, $f_{\rho, \mathcal{B}}(z \mid x, y)$ is guaranteed to be maximized at the correct logit $z = x \star y$.[6] We find that each trained model implements several such circuits and that the logits are approximately a linear combinations of terms of the form Equation 2.

We now explain how the model weights implement the circuit in Eq 2. Recall from Eq 1 that $f_{\boldsymbol{\theta}}$ can be written as the sum of the contributions of each neuron plus the bias. As seen in Section 6.2, each neuron is in the span of a single irrep $\rho \in \text{Irrep}(G)$. We find that, furthermore, these neurons

---

[4]The term "$\rho$-set" is our own. All other definitions and notations introduced in this section are standard.

[5]To see the equivariance, note that if $f_{\rho, \mathcal{B}}$ is of this form, then, for $g, h \in G$, we have $f_{\rho, \mathcal{B}}(z \mid g \star x, y \star h) = f_{\rho, \mathcal{B}}(g^{-1} \star z \star h^{-1} \mid x, y)$.

[6]Notice that $z = x \star y$ if and only if $x^{-1} \star z \star y^{-1} = e$, which implies $\rho(x^{-1} \star z \star y^{-1}) = \boldsymbol{I}$. This implication goes both ways if $\rho$ is faithful, which is the case for all irreps of $S_5$ with dimension $> 1$.

are actions of $\rho$ on finite $\rho$-sets $\mathcal{B} \subseteq \mathbb{R}^d$ projected onto one dimension. Moreover, the unembedding weights of each neuron are related to the left and right embedding weights, so that, for some $\boldsymbol{b}, \boldsymbol{b}'$ from a $\rho$-set $\mathcal{B}$ and some projection vector $\boldsymbol{a}$, up to scaling,

$$\boldsymbol{w}_u^i(z) = -\boldsymbol{b}^\top \rho(z)\boldsymbol{b}', \qquad \boldsymbol{w}_l^i(x) = \boldsymbol{b}^\top \rho(x)\boldsymbol{a}, \qquad \boldsymbol{w}_r^i(y) = -\boldsymbol{a}^\top \rho(y)\boldsymbol{b}'. \qquad (3)$$

Additionally, there is one neuron of this form for each of the $|\mathcal{B}|^2$ pairs $(\boldsymbol{b}, \boldsymbol{b}')$,[7] and $\boldsymbol{a}$ is constant across all such pairs. See Observation B.1 for a full enumeration of our findings.

Based on these observations, we partition neurons into independent $\rho$-*set circuits*; each circuit is associated with a $\rho \in \mathrm{Irrep}(G)$ of dimension $d$, a finite $\rho$-*set* $\mathcal{B} \subseteq \mathbb{R}^d$ that $\rho$ acts on transitively by permutation, and a constant vector $\boldsymbol{a} \in \mathbb{R}^d$. Call this circuit $f_{\rho, \mathcal{B}}$; then,

$$f_{\rho, \mathcal{B}}(z \mid x, y) = - \sum_{\boldsymbol{b}, \boldsymbol{b}' \in \mathcal{B}} \boldsymbol{b}^\top \rho(z)\boldsymbol{b}' \, \mathrm{ReLU}[\boldsymbol{b}^\top \rho(x)\boldsymbol{a} - \boldsymbol{a}^\top \rho(y)\boldsymbol{b}'] \qquad (4)$$

Using the $\rho$-set structure of $\mathcal{B}$, we can change variables via $\tilde{\boldsymbol{b}} = \rho(x)^\top \boldsymbol{b} = \rho(x^{-1})\boldsymbol{b}$ and $\tilde{\boldsymbol{b}}' = \rho(y)\boldsymbol{b}'$, and arrive at the aforementioned circuit Eq 2 (with $\boldsymbol{b}, \boldsymbol{b}'$ exchanged for $\tilde{\boldsymbol{b}}, \tilde{\boldsymbol{b}}'$). Since $f_{\boldsymbol{\theta}}$ is a sum of such bi-equivariant circuits, it is also bi-equivariant.

Further, terms of the summation in Eq 2 are zero whenever $\boldsymbol{b} = \boldsymbol{b}'$, which is precisely when $\boldsymbol{b}^\top \rho(e)\boldsymbol{b}' = \langle \boldsymbol{b}, \boldsymbol{b}' \rangle$ is largest. Heuristically, these properties explain why $f_{\rho, \mathcal{B}}(z \mid x, y)$ is maximized when $z = x \star y$. For certain $\rho \in \mathrm{Irrep}(G)$, we can rigorously show that the function is indeed maximized at $z = x \star y$ for any value of $\boldsymbol{a}$; see Lemma G.5 for details. Eq 2 can also be as a separating hyperplane in the ambient space inhabited by irreps $\rho$. See Sec B.2 for details.

## 4.2 THE SIGN CIRCUIT

For irreps of dimension strictly greater than one, we observe that the model learns circuits closely approximating what we describe in Section 4.1. However, for the sign irrep,[8] the model is able to use one-dimensionality to avoid the expense of the double summation in Eq 4.

Explicitly, up to scaling, the sign circuit is

$$f_{\mathrm{sgn}}(z \mid x, y) = \rho(z) - \rho(z)\,\mathrm{ReLU}[\rho(x) - \rho(y)] - \rho(z)\,\mathrm{ReLU}[\rho(y) - \rho(x)] = \rho(x^{-1} \star z \star y^{-1}),$$

where $\rho$ is the sign irrep. The circuit comprises two neurons $-\rho(z)\,\mathrm{ReLU}[\rho(x) - \rho(y)]$ and $-\rho(z)\,\mathrm{ReLU}[\rho(y) - \rho(x)]$, and uses the unembedding bias to strip out an extraneous $\rho(z)$ term. See Appendix B.5 for further discussion.

## 5 EXPLANATIONS AS COMPACT PROOFS OF MODEL PERFORMANCE

Following Gross et al. (2024), we evaluate the completeness of a mechanistic explanation by translating it into a compact proof of model performance. Intuitively, given the model weights and an interpretation of the model, we aim to leverage the interpretation to efficiently compute a guarantee on the model's global accuracy. More precisely, we construct a *verifier* program[9] $V$ that takes as input the model parameters $\boldsymbol{\theta}$, the group $G$, and an encoding of the interpretation into a string $\pi$, and returns a real number. We require that $V$ always provides valid lower bounds on accuracy; that is, $V(\boldsymbol{\theta}, G, \pi) \leq \alpha_G(\boldsymbol{\theta})$ regardless of the given interpretation $\pi$. [10] Our measure of $\pi$'s faithfulness is the tightness of the output guarantee $V(\boldsymbol{\theta}, G, \pi)$, i.e. how close it is true to the true accuracy $\alpha_G(\boldsymbol{\theta})$.

Two simple examples of verifiers are:

---

[7]To be more precise, we find that there are often multiple neurons per $(\boldsymbol{b}, \boldsymbol{b}')$. However, they are scaled such that the sum of neuron contributions corresponding to each pair is uniform.

[8]The sign irrep maps permutations to $\pm 1$ depending on whether they decompose into an odd or even number of transpositions. It is the only nontrivial one-dimensional irrep of $S_n$. In general, any real-valued one-dimensional irrep of any group must take values $\pm 1$, and the same circuit as described here works.

[9]Formally, a Turing machine. We elide any implementation details related to encoding finite group-theoretic objects as strings, error from finite floating-point precision, etc.

[10]This *soundness* requirement prevents $\pi$ from (for example) simply providing the true accuracy $\alpha_G(\boldsymbol{\theta})$ to the verifier. If $V$ takes $\pi$'s veracity for granted, it is no longer sound—$\pi$ could falsely claim the accuracy to be higher than the truth, causing $V(\boldsymbol{\theta}, G, \pi) > \alpha_G(\boldsymbol{\theta})$.

1. The vacuous verifier $V_{\text{vac}}(\boldsymbol{\theta}, G, \emptyset) = 0$. This is a valid verifier because 0 is a lower bound on any model's accuracy.

2. The brute force verifier $V_{\text{brute}}(\boldsymbol{\theta}, G, \emptyset) = \alpha_{\text{G}}(\boldsymbol{\theta})$, which takes the model's weights and runs its forward pass on every input to compute the global accuracy.

While the brute force approach attains the optimal bound by recovering the true accuracy, it is computationally expensive. (Indeed, it is intractable in any real-world setting, where the input space is too large to enumerate.) Notice also that neither example is provided an interpretation $\pi$; without any information about $\boldsymbol{\theta}$, we cannot expect the verifier to do better than these trivial examples. We aim to construct verifiers that, when $\pi$ is a meaningful interpretation, (1) give *non-vacuous* guarantees on model accuracy and (2) are *compact*, i.e. more time-efficient than brute force.

A good understanding of the model's internals should allow us to compress its description and therefore make reasonable estimates on its error. A more complete explanation should yield a tighter bound, while also reducing the computation cost. Therefore if we think of bounding the accuracy as a trade-off between accuracy and computational cost, a good explanation of the model should push the Pareto frontier outward. See Gross et al. (2024) for a more thorough discussion.

## 5.1 Constructing compact proofs

The verifier lower-bounds the model's accuracy by trying to prove for each $x, y \in G$ that the model's output is maximized at $x \star y$, i.e. that $f_{\boldsymbol{\theta}}(x \star y \mid x, y) > \max_{z \neq x \star y} f_{\boldsymbol{\theta}}(z \mid x, y)$. More precisely, the verifier strategy is:

(1) Given model parameters $\boldsymbol{\theta}$, use the interpretation $\pi$ to construct an *idealized model* $\tilde{\boldsymbol{\theta}}$.

(2) For $x, y \in G$, lower bound the *margin* $f_{\tilde{\boldsymbol{\theta}}}(x \star y \mid x, y) - \max_{z \neq x \star y} f_{\tilde{\boldsymbol{\theta}}}(z \mid x, y)$.

(3) For $x, y \in G$, upper bound the *maximum logit distance*

$$\max_{z \neq x \star y} |f_{\tilde{\boldsymbol{\theta}}}(z \mid x, y) - f_{\boldsymbol{\theta}}(z \mid x, y)| + f_{\tilde{\boldsymbol{\theta}}}(x \star y \mid x, y) - f_{\boldsymbol{\theta}}(x \star y \mid x, y).$$

(4) The accuracy lower bound is the proportion of inputs $x, y \in G$ such that the margin lower bound exceeds the distance upper bound. For such input pairs, the margin by which the idealized model's logit value on the correct answer exceeds the logit value of any incorrect answer is larger than the error between the original and idealized model, so the original model's logit output must be maximized at the correct answer as well. See Figure 2.

Recall our model architecture Eq 1. The brute-force verifier runs a forward pass with time complexity $O(m|G|)$ over $|G|^2$ input pairs, and so takes time $O(m|G|^3)$ total. [11] Our verifiers need to be asymptotically faster than this in order to be performing meaningful compression. Naïvely, though, both steps (2) and (3) take time $O(m|G|^3)$, no better than brute force. However, we can reduce the time complexity of each to $O(m|G|^2)$: for (2) by exploiting the internal structure of the idealized model, and for (3) by using Lemma G.1.

**Compact proofs via coset concentration** Intuitively, coset concentration (Stander et al., 2024) gives a way to perform nontrivial compression—if the interpretation says that a neuron is constant on the cosets of a specific group, then we need only to check one element per coset, instead of a full iteration over $G$. However, the shortcomings listed in Section 6.1 are an obstacle to formalizing this intuition into a compact proof. The verifier $V_{\text{coset}}$ we construct pessimizes over the degrees of freedom not explained by the coset concentration explanation, resulting in accuracy bounds that are vacuous (Section 5.2). See Appendix D for details of $V_{\text{coset}}$'s construction.

**Compact proofs via $\rho$-sets** We are able to turn our $\rho$-set circuit interpretation into a proof of model accuracy that gives non-vacuous results on a majority of trained models; see Appendix E for details and Section 5.2 for empirical results.

---

[11]For simplicity of presentation, we assume all matrix multiplication is performed with the naïve algorithm; that is, the time complexity of multiplying two matrices of size $m \times n$ and $n \times k$ is $O(mnk)$.

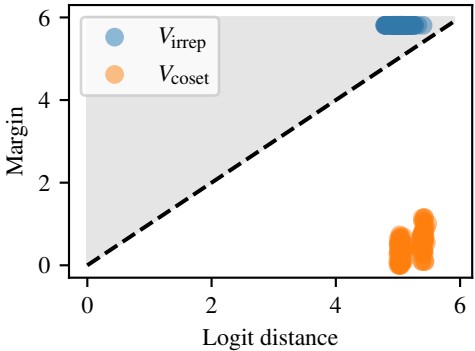

The interpretation string $\pi$ labels each neuron with its corresponding irrep $\rho$ and its $\rho$-set. The verifier $V_{\mathrm{irrep}}$ is then able to use this interpretation string to construct an idealized version of the input model that implements $\rho$-set circuits (Eq 4) exactly. By bi-equivariance, this idealized model's accuracy can then be checked with a **single forward pass**. Finally, $V_{\mathrm{irrep}}$ bounds the distance between the original and the idealized models using Lemma G.1.

Figure 2: Margin lower bound vs. logit distance upper bound over $x, y \in S_5$ for $V_{\mathrm{irrep}}$ and $V_{\mathrm{coset}}$ on a single example model. The accuracy lower bound is precisely the number of points for which the margin lower bound is larger than the logit upper bound (shaded region); in this example, the bound from $V_{\mathrm{irrep}}$ is 100% while that from $V_{\mathrm{coset}}$ is 0%. The margin lower bound of $V_{\mathrm{irrep}}$ is constant due to bi-equivariance.

## 5.2 EMPIRICAL RESULTS FOR COMPACT PROOFS

We train 100 one-hidden-layer neural network models from random initialization on the group $S_5$. We then compute lower bounds on accuracy obtained by brute force, the cosets explanation, and the $\rho$-sets explation, which we refer to as $V_{\mathrm{brute}}, V_{\mathrm{coset}}, V_{\mathrm{irrep}}$ respectively. We evaluate these lower bounds on both their runtime[12] (compactness) and by the tightness of the bound.

See Appendix I for full experiment details.

As expected, $V_{\mathrm{brute}}$ obtains the best accuracy bounds (indeed, they are exact), but has a slower runtime than $V_{\mathrm{irrep}}$; see Figure 3. On the other hand, across all experiments, $V_{\mathrm{coset}}$ failed to yield non-vacuous accuracy bounds; while the margin lower bounds of $V_{\mathrm{coset}}$'s idealized models are nonzero, they are swamped by the upper bound on logit distance to the original model; see Figure 2.

Looking again at Figure 3, we see that the accuracy bound due to $\rho$-sets is bimodal: the verifier $V_{\mathrm{irrep}}$ obtains a bound of near 100% for roughly half the models, and a vacuous bound of 0% for another half. Investigating the models for which $V_{\mathrm{irrep}}$ does not obtain a good bound, we are able to discover for many of them aspects in which they deviate from our $\rho$-sets explanation:

- ($\boldsymbol{a}$-bad) The $\boldsymbol{a}$ projection vector (Eq 4) fails to be constant across terms of the double sum over $\boldsymbol{b}, \boldsymbol{b}' \in \mathcal{B}$, so the change of variables showing bi-equivariance (Eq 2) is invalid.[13] We find that such models have poorer cross-entropy loss and larger weight norm than those with constant $\boldsymbol{a}$, suggesting they have converged to a suboptimal local minimum (Figure 4 in Appendix B.6).

- ($\rho$-bad) The double sum over $\mathcal{B}$ (Eq 4) misses some $\boldsymbol{b}, \boldsymbol{b}'$ pairs. Again, we are unable to prove bi-equivariance when this happens. We speculate that in this case the model is approximating the discrete summation by a numerical integral à la Yip et al. (2024).

Although these cases are failures of our $\rho$-sets interpretation to explain the model, their presence can be seen as a success for compact proofs as a measure of interpretation completeness. **For models we do not genuinely understand, we are unable to achieve non-vacuous guarantees on accuracy.**

## 6 REVISITING PREVIOUS EXPLANATIONS: COSETS AND IRREPS

In this section, we recall the coset algorithm described in Stander et al. (2024), and the notion of irrep sparsity observed in Chughtai et al. (2023). We find that although these works correctly identify properties of individual neuron weights, they lack a precise picture of how these neurons combine to compute the group operation. We conclude the section by clarifying the logical relationship between these observations and our present work.

---

[12]We use the verifier's time elapsed as a proxy for FLOPs, which is a non-asymptotic measure of runtime.

[13]If $\boldsymbol{a}$ depends on $(\boldsymbol{b}, \boldsymbol{b}')$, then there is a remaining dependence on $(\rho(x)\tilde{\boldsymbol{b}}, \rho(y^{-1})\tilde{\boldsymbol{b}}')$ inside the ReLU after changing variables.

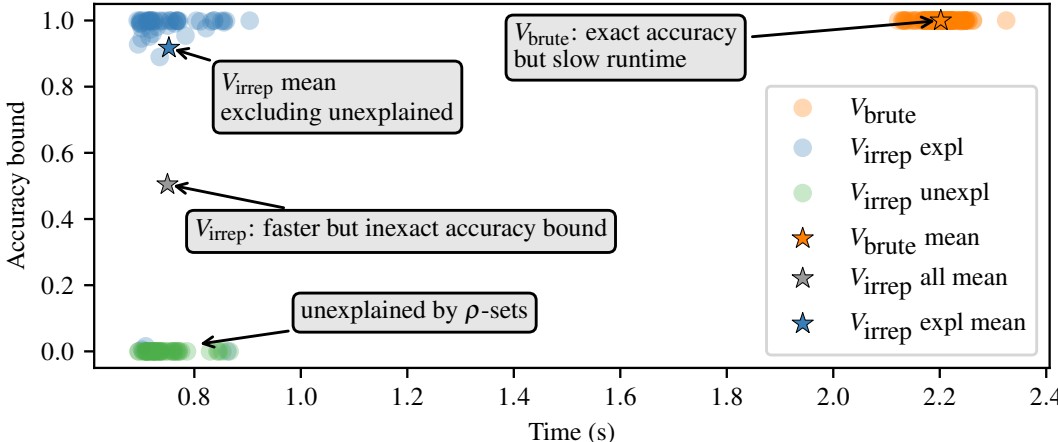

Figure 3: Accuracy bound vs. computation time for $V_{\mathrm{irrep}}$ and $V_{\mathrm{brute}}$ on 100 models trained on $S_5$. Points in **green** ($V_{\mathrm{irrep}}$ unexpl) are models for which we find by inspection that our $\rho$-sets explanation does not hold, i.e. either ($a$-bad) or ($\rho$-bad). Mean accuracy bound is 100% for $V_{\mathrm{brute}}$ (**orange**), 0% for $V_{\mathrm{coset}}$ (not shown), 50.4% for $V_{\mathrm{irrep}}$ (union of **blue** and **green**), and 91.7% for $V_{\mathrm{irrep}}$ when only including models for which neither ($a$-bad) nor ($\rho$-bad) occur (**blue**, 55% of total). Mean time elapsed is 2.20s for $V_{\mathrm{brute}}$ and 0.75s for $V_{\mathrm{irrep}}$. The asymptotic time complexity of $V_{\mathrm{brute}}$ is $O(m|G|^3)$ while that of $V_{\mathrm{irrep}}$ is $O(m|G|^2)$.

## 6.1 COSET CONCENTRATION

Recall the model architecture Eq 1. In Stander et al. (2024), they make the following observation: For each neuron $i$, the left embeddings are approximately constant on the *right* cosets of a certain subgroup $K_1$ of $G$, while the right embeddings are approximately constant on the *left* cosets of a conjugate subgroup $K_2 = g^{-1}Kg$. That is, they are coset concentrated; see Definition 6.2. They then observe that there is a subset $X = K_1 \star h = h' \star K_2$. On inputs $x, y \in G$, the left and right embeddings of the neuron $i$ sum to near zero precisely when $x \star y \in X$. Meanwhile, the unembedding takes smaller values on elements of $X$; thus, when $x \star y \notin X$, the model's confidence in $G \setminus X$ is increased.[14] In the example of $S_5$, these $X$ typically take the form of $X_{ab} := \{\sigma \in S_5 \mid \sigma(a) = b\}$. See Stander et al. (2024, Section 5) and Appendix A for more details.

This explanation leaves several things unclear:

- Even given coset concentration, there are many choices of left/right embeddings that sum to zero whenever $x \star y \in X$. The choice matters: for example, if there are many input pairs where $x \star y \notin X$ but the sum of embeddings is near zero, then the model cannot be expected to perform well. How do we know whether the model has made a good choice?
- The unembedding is similarly underdetermined: there are many degrees of freedom in choosing weights satisfying the sole constraint of being smaller on $X$ than on $G \setminus X$.
- The bias term is not mentioned, despite being present in the models under study.

In Proposition A.2, we provide a more precise version of this explanation, assuming the model weights satisfy stronger constraints. However, the actual models we investigate do not match these assumptions closely, which is reflected by our failure to convert this explanation into non-vacuous bounds in Appendix D.

## 6.2 IRREP SPARSITY

Chughtai et al. (2023) notice that each neuron in the trained model is in the linear span of matrix elements of some irrep $\rho \in \mathrm{Irrep}(G)$; we refer to this condition as *irrep sparsity* (see Definition 6.1 for a formal statement). Based on this observation, they propose that the model:

---

[14]These subgroups might vary from neuron to neuron

1. Embeds the input pair $x, y \in G$ as $d \times d$ irrep matrices $\rho(x)$ and $\rho(y)$.

2. Uses ReLU nonlinearities to compute the matrix multiplication $\rho(x)\rho(y) = \rho(x \star y)$.

3. Uses ReLU nonlinearities and $\rho(x \star y)$ from the previous step to compute $\mathrm{tr}(\rho(x \star y \star z^{-1}))$, which is maximized at $x \star y \star z^{-1} = e \iff z = x \star y$.

However, because they leave the ReLU computations as a black box, they are unable to fully explain the model's implemented algorithm. We were able to deduce a more complete description by carefully investigating *which* linear combinations of $\rho$ each neuron uses.

### 6.3 RELATING IRREP SPARSITY, COSET CONCENTRATION, AND $\rho$-SETS

This paper and prior works observe multiple properties of neurons viewed as functions $G \to \mathbb{R}$:

**Definition 6.1** (Chughtai et al. 2023)**.** A function $f \colon G \to \mathbb{R}$ is *irrep sparse* if it is a linear combination of the matrix entries of an irrep of $G$. That is, there exists a $\rho \in \mathrm{Irrep}(G)$ of dimension $d$ and a matrix $\boldsymbol{A} \in \mathbb{R}^{d \times d}$ such that $f(g) = \mathrm{tr}(\rho(g)\boldsymbol{A})$.

**Definition 6.2** (Stander et al. 2024)**.** A function $f \colon G \to \mathbb{R}$ is *coset concentrated* if there exists a nontrivial subgroup $H \leq G$ such that $f$ is constant on the cosets (either left or right) of $G$.

**Definition 6.3.** A function $f \colon G \to \mathbb{R}$ is a *projected $\rho$-set* for $\rho \in \mathrm{Irrep}(G)$ of dimension $d$ if there exist $\boldsymbol{a}, \boldsymbol{b} \in \mathbb{R}^d$ such that $f(g) = \boldsymbol{a}^\top \rho(g)\boldsymbol{b}$ and $\boldsymbol{b}$ is in a $\rho$-set with nontrivial stabilizer.

In this section, we clarify the logical relationships between these three properties.

**Projected $\rho$-sets are irrep-sparse** This fact, while immediate from definitions, resolves a mystery from Chughtai et al. (2023, Figure 7): why is the standard irrep $\rho_{\mathrm{std}}$ of $S_5$ learned significantly more frequently than the sign-standard irrep $\rho_{\mathrm{sgnstd}}$, when both have the same dimensionality $\dim \rho_{\mathrm{std}} = \dim \rho_{\mathrm{sgnstd}} = 4$? The answer is that the smallest $\rho_{\mathrm{std}}$-set has size 5, while the smallest $\rho_{\mathrm{sgnstd}}$-set has size 10 (Appendix H). Thus a minimum complete $\rho_{\mathrm{std}}$-set circuit needs $5^2 = 25$ neurons, while a minimum complete $\rho_{\mathrm{sgnstd}}$-set circuit needs $10^2 = 100$. The order of frequencies with which $\rho \in \mathrm{Irrep}(G)$ is learned (Chughtai et al., 2023, Figure 7) is the same as the ordering of $\mathrm{Irrep}(G)$ **by minimum $\rho$-set size** (Table 2), **not by dimensionality.**

**Projected $\rho$-sets are coset concentrated** Our $\rho$-set interpretation immediately explains coset concentration: the map $g \mapsto \rho(g)\boldsymbol{b}$ is constant on precisely the cosets of the stabilizer $\mathrm{Stab}_G(\boldsymbol{b}) := \{g \in G \mid \rho(g)\boldsymbol{b} = \boldsymbol{b}\}$. Further, since the action of $G$ is transitive, $\mathrm{Stab}_G(\boldsymbol{b}')$ for any other element $\boldsymbol{b}'$ is a conjugate of $\mathrm{Stab}_G(\boldsymbol{b})$; as a consequence left and right preactivations of each neuron concentrate on cosets of conjugate subgroups.

**Coset concentration fails to explain irrep sparsity** Stander et al. (2024) partially explain irrep sparsity via coset concentration. We paraphrase their key lemma here:

**Lemma 6.4** (Stander et al., 2024)**.** *Let $H$ be a subgroup of $G$. The Fourier transform of a function constant on the cosets of $H$ is nonzero only at the irreducible components of the permutation representation corresponding to the action of $G$ on $G/H$.*

This lemma fails to fully explain irrep sparsity, since the permutation representation of $G$ on $G/H$ is never itself an irrep; it always contains the trivial irrep, possibly among others. Thus, it does not explain why the models' neurons are supported purely on single irreps and not, say, on linear combinations of each irrep with the trivial irrep.

Notice also that, since there are many more subgroups than irreps (Stander et al., 2024, Appendix G.3), most subgroups $H \leq G$ have corresponding actions of $G$ on $G/H$ that decompose into more than two irreps; otherwise, since the decomposition is unique, there would be at most $|\mathrm{Irrep}(G)|^2$ total possibilities. Explicit examples of subgroups whose indicators are supported on more than two irreps are given in Appendix H.

**Projected $\rho$-sets are equivalent to the conjunction of irrep sparsity and coset concentration** Note that neither irrep sparsity nor coset concentration alone is equivalent to the condition of being a projected $\rho$-set. For an example of an irrep-sparse function that is not a $\rho$-set, consider the

function $\chi(g) = \mathrm{tr}(\rho(g))$. If $\rho$ is faithful and has nontrivial stabilizer (for example, the standard 4-dimensional irrep of $S_5$), then $\chi$ cannot be a projected $\rho$-set, because it is maximized uniquely at the identity (Chughtai et al., 2023, Theorem D.7) and thus is not coset-concentrated. To see that coset concentration does not imply being a projected $\rho$-set, recall from above that there are coset-concentrated functions that are not irrep-sparse, but all projected $\rho$-sets are irrep sparse.

On the other hand, if $f \colon G \to \mathbb{R}$ is both irrep-sparse and coset-concentrated, then it must be a projected $\rho$-set; for a proof see Lemma G.6. Hence, if we consider by itself a single embedding of a single neuron on $f_{\boldsymbol{\theta}}$, then our Observation B.2(2) is logically equivalent to the combination of observations from previous works. However, our perspective gives us more insight into the relationship between left/right embedding and unembedding neurons (Observation B.2(1)) as well as the relationship between different neurons (Observation B.2(6,7)).

## 7 Discussion

**Limitations of the $\rho$-sets interpretation**   The $\rho$-set explanation we provide has a rather limited scope: we only claim to understand roughly half of the models we examine, all of which are trained on $S_5$. On the other hand, all of the models we examine satisfy both irrep sparsity and coset concentration. It was by trying and failing to obtain non-vacuous compact guarantees on model accuracy that we discovered that our understanding of some models is still incomplete. Hence, compact proofs are a quantitative means of detecting gaps in proposed model explanations.

**Causal interventions**   In an attempt to verify the validity of the coset concentration interpretation, Stander et al. (2024) perform a series of causal interventions. While the results do not yield evidence that their interpretation is incorrect, they also do not provide strong evidence that it is correct. Indeed, we perform the same interventions on a model for which we know the cosets explanation cannot hold, and find results in the same direction; see Appendix F. Thus, in this case, causal interventions might yield strong negative evidence against an explanation, but provide weaker positive evidence. Furthermore, causal interventions lack a notion of an explanation's simplicity independent from its faithfulness—they do not provide a quantitative measure of how much an explanation compresses the model.

**Compact proofs**   For the models we do understand, we obtain tight accuracy bounds in significantly less time than brute force, providing strong positive evidence for our understanding. Hence, we view the compact proof approach as complementary to the causal intervention one: **A tight bound is strong positive evidence for an explanation, whereas a poor or vacuous bound does not give us strong negative evidence.** Indeed, the coset interpretation does make nontrivial observations and yield partial explanations of model performance, but this is not reflected in the vacuous bounds we obtain. It may be the case that some of the pessimizations we use in our construction of $V_{\mathrm{coset}}$ were unnecessarily strong; the translation from an informal explanation into a rigorous bound is itself informal, by necessity.

## 8 Conclusion

Multiple previous works (Chughtai et al., 2023; Stander et al., 2024) have examined the group composition setting and claimed a mechanistic understanding of model internals. However, as we have demonstrated here, these works left much internal structure unrevealed. Our own interpretation incorporates this structure, resulting in a more complete explanation that unifies previous attempts.

We verify our explanation with compact proofs of model performance, and obtain strong positive evidence that it holds for a large fraction of the models we investigate. For models where we fail to obtain bounds, we find that many indeed do not fit our proposed interpretation. Compact proofs thus provide rigorous and quantitative positive evidence for an interpretation's completeness. We see this work as a preliminary step towards a more rigorous science of interpretability.

## REPRODUCIBILITY STATEMENT

See Appendix I for experiment details. Code for reproducing our experiments can be found at `https://anonymous.4open.science/r/groups-E024`.

## ACKNOWLEDGEMENTS

We thank Jesse Hoogland, Daniel Filan, Ronak Mehta, Mathijs Henquet, and Alice Rigg for helpful discussions and feedback on drafts of this paper. WW was supported by a grant from the Long-Term Future Fund. JD was supported by a grant from Open Philanthropy. This research was conducted in part during the ML Alignment & Theory Scholars Program. Computing resources for some experiments were provided by Timaeus. We thank the anonymous reviewers for providing many suggestions that improved the clarity of this paper.

## AUTHOR CONTRIBUTIONS

**Wilson Wu**   Led the engineering and ran the bulk of all experiments. Designed, implemented, and ran compact proofs. Proved majority of the theoretical results. Contributed to writing of paper and rebuttals.

**Louis Jaburi**   Initiated the project and suggested experiments to run. Contributed to writing of paper and rebuttals.

**Jacob Drori**   Made contributions leading to discovery of the $\rho$-set circuit: pushed to study the structure of $A_i, B_i, C_i$ (defined in Observation B.1), predicted Observation 1; predicted $\{b_i\}$ is a $\rho$-set; noticed $b_i$ and $c_i$ vary independently, leading to the double-sum in Equation 4.

**Jason Gross**   Advised the project, including developing proof strategies and providing feedback on experimental results and presentation.

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

## A COSET CONCENTRATION IN DETAIL

The main observation of Stander et al. (2024) is that neurons are concentrated on the cosets of subgroups. More precisely,

**Observation A.1** (Stander et al., 2024). *For each $i \in [m]$, there exists a subgroup $H_i \leq G$ and element $g_i \in G$ such that the left and right neurons are approximately constant on the right cosets of $H_i$ and the left cosets of the conjugate subgroup $K_i = g_i H_i g_i^{-1}$,[15] respectively:*

$$xy^{-1} \in H_i \implies \boldsymbol{w}_l^i(x) \approx \boldsymbol{w}_l^i(y), \qquad x^{-1}y \in K_i^{-1} \implies \boldsymbol{w}_r^i(x) \approx \boldsymbol{w}_r^i(y). \tag{5}$$

*This forces the summed embeddings $\boldsymbol{w}_l^i(x) + \boldsymbol{w}_r^i(y)$ to be approximately constant on the double cosets $H_i \backslash G / K_i$. Further, it is observed that*

$$\boldsymbol{w}_l^i(x) + \boldsymbol{w}_r^i(y) \approx 0 \iff xy \in H_i x y g_i K_i = H_i g_i^{-1}. \tag{6}$$

If we include several additional assumptions, this observation is sufficient for $f_{\boldsymbol{\theta}}$ to attain perfect accuracy. Note it was not explicitly investigated in Stander et al. (2024) to what extent these additional conditions are met.

**Proposition A.2.** *Suppose the model parameters $\boldsymbol{\theta}$ are such that Observation A.1 holds, the unembeddings satisfy*

$$\max_{z \notin H_i g_i^{-1}} \boldsymbol{w}_u^i(z) = \min_{z \notin H_i g_i^{-1}} \boldsymbol{w}_u^i(z) > \max_{z \in H_i g_i^{-1}} \boldsymbol{w}_u^i(z) \tag{7}$$

---

[15]Throughout the appendix, we denote the group multiplication of $x, y \in G$ by $xy$ instead of by $x \star y$.

and the bias $\boldsymbol{w}_b$ is zero. Further, defining

$$s(H, g) = \min_{\substack{x,y \in G \\ xy \notin H_i g_i^{-1}}} \sum_{\substack{i \in [m] \\ (H_i, g_i) = (H, g)}} \text{ReLU}[\boldsymbol{w}_l^i(x) + \boldsymbol{w}_r^i(y)], \tag{8}$$

suppose every singleton $\{z\}$ for $z \in G$ can be written as an intersection of sets from the family

$$\{G - H_i g_i^{-1} \mid i \in [m], s(H_i, g_i) > 0\}$$

Then, $\alpha_{\mathrm{G}}(f) = 1$.

*Proof.* Let $x, y, z \in G$ with $z \neq xy$. Then,

$$\begin{aligned}
f(xy \mid x, y) - f(z \mid x, y) &= \sum_{i=1}^{m} (\boldsymbol{w}_u^i(xy) - \boldsymbol{w}_u^i(z)) \, \text{ReLU}[\boldsymbol{w}_l^i(x) + \boldsymbol{w}_r^i(y)] \\
&\geq \sum_{(H,g)} s(H,g) \mathbf{1}\{xy \notin H_i g_i^{-1}\} \sum_{\substack{i \in [m] \\ (H_i, g_i) = (H, g)}} (\boldsymbol{w}_u^i(xy) - \boldsymbol{w}_u^i(z)) \\
&\geq 0,
\end{aligned}$$

To see that the inequality is strict, choose $i$ such that $s(H_i, g_i) > 0$ with $z \in H_i g_i^{-1}$ and $xy \notin H_i g_i^{-1}$. $\qquad \square$

## B $\rho$-SET CIRCUITS IN DETAIL

### B.1 LIST OF NOVEL OBSERVATIONS

Chughtai et al. (2023) observe that neurons are irrep-sparse. That is,

**Observation B.1** (Chughtai et al., 2023). *For each $i \in [m]$, there exists a real-valued irrep $\rho_i \colon G \to \text{GL}(d_i, \mathbb{R})$ of degree $d_i$ as well as $\boldsymbol{A}_i, \boldsymbol{B}_i, \boldsymbol{C}_i \in \mathbb{R}^{d_i \times d_i}$ such that*

$$\boldsymbol{w}_l^i(x) \approx \text{tr}(\rho(x) \boldsymbol{A}_i), \quad \boldsymbol{w}_r^i(x) \approx \text{tr}(\rho(x) \boldsymbol{B}_i), \quad \boldsymbol{w}_u^i(x) \approx \text{tr}(\rho(x) \boldsymbol{C}_i).$$

However, this is not sufficient to fully describe how the model computes the group operation; in particular the ReLU nonlinearities are left as black boxes.

We observe further structure:

**Observation B.2** (Ours). *Let $(\rho_i, \boldsymbol{A}_i, \boldsymbol{B}_i, \boldsymbol{C}_i)_{i=1}^m$ be as in Observation B.1. Suppose that all irreps of $G$ over $\mathbb{C}$ are real-valued; in particular, this is the case for $S_n$[16]*

1. *$\boldsymbol{C}_i \approx r_i \boldsymbol{B}_i \boldsymbol{A}_i$ for some $r_i > 0$.*

2. *Each of $\boldsymbol{A}_i, \boldsymbol{B}_i, \boldsymbol{C}_i$ are rank one, with $\|\boldsymbol{A}_i\| \approx \|\boldsymbol{B}_i\|_F$. Hence, we may write*

$$\boldsymbol{A}_i \approx s_i \boldsymbol{a}_i \boldsymbol{b}_i^\top, \quad \boldsymbol{B}_i \approx s_i \boldsymbol{c}_i \boldsymbol{d}_i^\top, \quad \boldsymbol{C}_i \approx r_i s_i^2 \langle \boldsymbol{d}_i, \boldsymbol{a}_i \rangle \boldsymbol{c}_i \boldsymbol{b}_i^\top, \tag{9}$$

   *where $s_i, r_i \in \mathbb{R}$ and $\boldsymbol{a}_i, \boldsymbol{b}_i, \boldsymbol{c}_i, \boldsymbol{d}_i \in \mathbb{R}^{d_i}$ are unit vectors.*

3. *$\boldsymbol{a}_i = \boldsymbol{d}_i$.*

*Further, fix an irrep $\rho$ of $G$ and consider $I_\rho = \{i \in [m] \mid \rho_i = \rho\}$, the subset of neurons supported on $\rho$. Then,*

4. *$\boldsymbol{a}_i$ is approximately constant on $I_\rho$. That is, there exists a unit vector $\boldsymbol{a}_\rho \in \mathbb{R}^{d_i}$ such that*

$$\forall i \in I_\rho : \boldsymbol{a}_i \approx \boldsymbol{d}_i \approx \boldsymbol{a}_\rho.$$

---

[16]Equivalently, the Frobenius-Schur indicator of each $\rho \in \text{Irrep}(G)$ is positive. We briefly consider complex and quaternionic irreps in Appendix K.2.

5. $\{\boldsymbol{b}_i\}_{i\in I_\rho} \approx \{-\boldsymbol{c}_i\}_{i\in I_\rho}$.

6. *Each* $\{\boldsymbol{b}_i\}_{i\in I_\rho}$ *can be partitioned into* $\{\mathcal{B}_{\rho,q}\}_q$ *such that, for each neuron* $i$, *the corresponding vectors* $\boldsymbol{b}_i$ *and* $-\boldsymbol{c}_i$ *must belong to the same partition. Further,* $\rho$ *acts on each partition by permutations; that is,* $\rho$ *induces a left G-set structure on every* $\mathcal{B}_{\rho,q}$. *We say that such* $\mathcal{B}_{\rho,q}$ *is a* $\rho$-**set**.

7. *If* $\dim\rho > 1$, *then, for each partition* $\mathcal{B}_{\rho,q}$, *there exists* $c_{\rho,q} \in \mathbb{R}$ *such that, for every pair* $\boldsymbol{b}, \boldsymbol{b}' \in \mathcal{B}_{\rho,q}$,

$$\sum_{i\in I_\rho} s_i^3 r_i \mathbf{1}\{\boldsymbol{b}_i = \boldsymbol{b}, \boldsymbol{c}_i = -\boldsymbol{b}'\} \approx c_{\rho,q}.$$

8. *If* $\dim\rho = 1$, *then, since we assume* $\rho$ *is real-valued, it must be either trivial or the sign irrep. We observe that the former never occurs. In the latter case,* $\mathcal{B}_\rho = \{\pm 1\}$, *and there exist* $c_+, c_- \in \mathbb{R}$ *such that*

$$c_+ \approx \sum_{i\in I_\rho} s_i^2 t_i^2 r_i \mathbf{1}\{\boldsymbol{b}_i = 1, \boldsymbol{b}_j = 1\} \approx \sum_{i\in I_\rho} s_i^2 t_i^2 r_i \mathbf{1}\{\boldsymbol{b}_i = -1, \boldsymbol{b}_j = -1\},$$

$$c_- \approx \sum_{i\in I_\rho} s_i^2 t_i^2 r_i \mathbf{1}\{\boldsymbol{b}_i = 1, \boldsymbol{b}_j = -1\} \approx \sum_{i\in I_\rho} s_i^2 t_i^2 r_i \mathbf{1}\{\boldsymbol{b}_i = -1, \boldsymbol{b}_j = 1\}.$$

9. *The bias* $\boldsymbol{w}_b$ *satisfies*

$$\boldsymbol{w}_b(z) \approx (c_- - c_+)\operatorname{sgn}(z).$$

## B.2 SEPERATING HYPERPLANE INTERPRETATION

Another way to express Eq 2 is as

$$f_{\rho,\mathcal{B}}(z \mid x, y) = -\left\langle \rho(x^{-1} \star z \star y^{-1}), \sum_{\boldsymbol{b},\boldsymbol{b}'\in\mathcal{B}} \operatorname{ReLU}[\boldsymbol{a}^\top(\boldsymbol{b} - \boldsymbol{b}')]\boldsymbol{b}'\boldsymbol{b}^\top \right\rangle, \tag{10}$$

where $\langle\cdot, \cdot\rangle$ is the Frobenius inner product of matrices.[17]. That is, each $\rho$-set circuit learns a $(\boldsymbol{a}, \mathcal{B})$-parameterized separating hyperplane $\langle\cdot, \boldsymbol{Z}(\boldsymbol{a}, \mathcal{B})\rangle$ between $\rho(e) = \boldsymbol{I}$ and $\{\rho(z) \mid z \neq e\}$. Since the $\rho$ are all unitary and thus of equal Frobenius norm, such a hyperplane always exists (e.g. $\boldsymbol{Z} = \boldsymbol{I}$), though we do not show in general that it can be expressed in the form of Eq 10.

## B.3 BI-EQUIVARIANCE

The observations in Section B.1 force the network to be *bi-equivariant*:

**Proposition B.3.** *If* $f_{\boldsymbol{\theta}}$ *satisfies Observations B.1 and B.2 exactly, then for all* $x, y, z, g_1, g_2 \in G$,

$$f_{\boldsymbol{\theta}}(z \mid g_1 x, y g_2) = f_{\boldsymbol{\theta}}(g_1^{-1} z g_2^{-1} \mid x, y).$$

*We say that* $f_{\boldsymbol{\theta}}$ *is* **bi-equivariant**. *In particular,*

$$f_{\boldsymbol{\theta}}(z \mid x, y) = f_{\boldsymbol{\theta}}(x^{-1} z y^{-1} \mid e, e),$$

*so such* $f_{\boldsymbol{\theta}}$ *depends only on* $x^{-1} z y^{-1}$. *Observe that* $x^{-1} z y^{-1} = e$ *iff* $z = xy$.

*Proof.* In the notation of Observation B.2, for each $\rho \in \operatorname{Irrep}(G)$ and partition $\mathcal{B}_{\rho,q}$, let $f_{\boldsymbol{\theta}}^{\rho,q} = \sum_{i\in I_\rho} \mathbf{1}\{\boldsymbol{b}_i \in \mathcal{B}_{\rho,q}\} f_{\boldsymbol{\theta}}^i$. Then, if $f_{\boldsymbol{\theta}}$ satisfies Observation B.2 exactly,

---
[17]$\langle\boldsymbol{A}, \boldsymbol{B}\rangle := \operatorname{tr}(\boldsymbol{A}^\top \boldsymbol{B}) = \sum_{i,j} \boldsymbol{A}_{i,j} \boldsymbol{B}_{i,j}$.

- If $\dim \rho > 1$, for every $\mathcal{B}_{\rho,q}$, there exists $\boldsymbol{a} \in \mathbb{R}^{\dim \rho}, c \in \mathbb{R}$ (all depending on $\rho$) such that, by re-indexing neurons,

$$f_{\boldsymbol{\theta}}^{\rho,q}(z \mid x, y) = -c \sum_{i,j=1}^{k} \boldsymbol{b}_i^\top \rho(z)\boldsymbol{b}_j \operatorname{ReLU}[\boldsymbol{b}_i^\top \rho(x)\boldsymbol{a} - \boldsymbol{a}\rho(y)\boldsymbol{b}_j]$$

$$= \sum_{i,j=1}^{k} \boldsymbol{b}_i^\top \rho(x^{-1}zy^{-1})\boldsymbol{b}_j \operatorname{ReLU}[\boldsymbol{a}^\top(\boldsymbol{b}_i - \boldsymbol{b}_j)]$$

$$= \left\langle \rho(x^{-1}zy^{-1}), -c \sum_{i,j}^{k} \operatorname{ReLU}[\boldsymbol{a}^\top(\boldsymbol{b}_i - \boldsymbol{b}_j)]\boldsymbol{b}_j\boldsymbol{b}_i^\top \right\rangle.$$

- If $\dim \rho = 1$, then $\rho$ must be the sign irrep. In this case,

$$\begin{aligned}
&f_{\boldsymbol{\theta}}^{\rho}(z \mid x, y) + \boldsymbol{w}_b(z) \\
&= c_+\rho(z)\operatorname{ReLU}[\rho(x) + \rho(y)] + c_+\rho(z)\operatorname{ReLU}[-\rho(y) - \rho(x)] - c_+\rho(z) \\
&\quad - c_-\rho(z)\operatorname{ReLU}[\rho(x) - \rho(y)] - c_-\rho(z)\operatorname{ReLU}[\rho(y) - \rho(x)] + c_-\rho(z) \\
&= c_+(\rho(z)|\rho(x) + \rho(y)| - \rho(z)) - c_-(\rho(z)|\rho(x) - \rho(y)| - \rho(z)) \\
&= c_+(\rho(z)(1 + \rho(xy)) - \rho(z)) - c_-(\rho(z)(1 - \rho(xy)) - \rho(z)) \\
&= (c_+ + c_-)\rho(zxy) \\
&= (c_+ + c_-)\rho(x^{-1}zy^{-1}).
\end{aligned}$$

$\square$

## B.4 STEPS TO DISCOVER THE $\rho$-SET CIRCUIT

In the main text, we presented the $\rho$-set circuit, and validated it by using our new understanding to efficiently lower-bound model performance. However, we did not describe the process by which we discovered the circuit in the first place. Here, we lead the reader through the steps of this process. This section can be viewed as an annotated walkthrough of the list of observations in Section B.1.

0. **Train a network on $G$-multiplication.** Recall that, given inputs $x, y \in G$, the trained network assigns the following logit value to output $z$:

$$f_{\boldsymbol{\theta}}(z \mid x, y) = \boldsymbol{w}_b(z) + \sum_i \boldsymbol{w}_u^i(x) \operatorname{ReLU}[\boldsymbol{w}_l^i(x) + \boldsymbol{w}_r^i(y)].$$

1. **Confirm neurons are irrep-sparse**. Given a $d$-dimensional irrep $\rho$ of $G$, construct[18] a tensor $T_\rho$ of shape $(|G|, d, d)$, interpreted as a $d \times d$ representation matrix for each element of $G$. We may also think of $T_\rho$ as $d^2$ many vectors of size $|G|$. We compute the span of these vectors $S_\rho = \operatorname{span}(T_\rho) \subseteq \mathbb{R}^{|G|}$. By the Schur orthogonality relations, the subspaces $\{S_\rho\}_{\rho \in \operatorname{Irrep}(G)}$ are mutually orthogonal.

    Now, if we fix a neuron $i$, then $\boldsymbol{w}_l^i, \boldsymbol{w}_r^i, \boldsymbol{w}_u^i$ are each functions $G \to \mathbb{R}$, i.e. vectors of dimensionality $|G|$. If these vectors are all approximately contained in a single $S_\rho$ (say each with $> 90\%$ variance explained), we say the neuron is supported on $\rho$. We then use least-squares linear regression to find $\boldsymbol{A}_i, \boldsymbol{B}_i, \boldsymbol{C}_i \in \mathbb{R}^{d \times d}$ satisfying:

$$\boldsymbol{w}_l^i(x) \approx \operatorname{tr}(\rho(x)\boldsymbol{A}_i), \quad \boldsymbol{w}_r^i(x) \approx \operatorname{tr}(\rho(x)\boldsymbol{B}_i), \quad \boldsymbol{w}_u^i(x) \approx \operatorname{tr}(\rho(x)\boldsymbol{C}_i).$$

    Note: the number of features in each of the three least-squares problems is $d^2$, and the number of data points is $|G| > d^2$. We can now write the network as

$$f_{\boldsymbol{\theta}}(z \mid x, y) \approx \sum_i \operatorname{tr}(\rho(z)\boldsymbol{C}_i) \operatorname{ReLU}[\operatorname{tr}(\rho(x)\boldsymbol{A}_i + \rho(y)\boldsymbol{B}_i)]$$

    and our task hereafter is to look for structure in $\boldsymbol{A}_i, \boldsymbol{B}_i, \boldsymbol{C}_i$.

---

[18]We use GAP (2024) for this. See `src/groups.py:get_real_irreps` in the provided repository.

2. **Observe $C_i \approx r_i B_i A_i$ for some $r_i > 0$.** We guess this relation by analogy with the modular addition circuit in Yip et al. (2024). We then verify it by defining Frobenius-normalized matrices

$$\hat{A}_i = A_i/\|A_i\|, \quad \hat{B}_i = B_i/\|B_i\|, \quad \hat{C}_i = C_i/\|C_i\|$$

and noting that the unexplained variance $\|\hat{C}_i - \hat{B}_i \hat{A}_i\|^2/\|\hat{C}_i\|^2$ is small.

3. **Observe that for real irreps, $A_i, B_i, C_i$ are approximately rank 1, with $\|A_i\| \approx \|B_i\|$.** For each matrix, we check that the variance explained by its top principal component is $\approx 1$. We also check $\|A_i\|/\|B_i\| \approx 1$. So we have

$$A_i \approx s_i a_i b_i^\top, \quad B_i \approx s_i c_i d_i^\top, \quad C_i \approx r_i s_i^2 \langle d_i, a_i \rangle c_i b_i^\top$$

where $s_i$ is the top singular value of $A_i$ or $B_i$, $(a_i, b_i)$ are the top left and right singular vectors of $A_i$, and $(c_i, d_i)$ are the top left and right singular values of $B_i$.

4. **Observe $a_i \approx d_i$.** We check the unexplained variance $\|a_i - d_i\|^2/\|a_i\|^2$ is small.

Now we restrict to neurons $i \in I_\rho$, i.e. those supported on a given irrep $\rho$. Let us take stock, and use our observations thus far to write out the contribution to the network due to these neurons:

$$f_{\boldsymbol{\theta}}^\rho(z \mid x, y) = \sum_{i \in I_\rho} r_i s_i^3 b_i^\top \rho(z) c_i \, \mathrm{ReLU}[b_i^\top \rho(x) a_i + a_i^\top \rho(y) c_i].$$

5. **Observe $a_i \approx a$, a constant.** We simply check $\langle a_i, a_j \rangle \approx 1$ for all pairs $i, j$. Now the only remaining degrees of freedom to understand are the $b_i$ and $c_i$.

6. **Cluster $\{b_i\}$ and $\{c_i\}$.** Using $k$-means clustering, [19] we find that $\{b_i\}$ and $\{c_i\}$ each consist of tight clusters. Let $\mathcal{B}_\rho$ and $\mathcal{C}_\rho$ be the sets of means of these respective clusters.

   In the simplest (and most illustrative) case, $\{(b_i, c_i)\}_{i \in I_\rho} = \mathcal{B}_\rho \times \mathcal{C}_\rho$. In other words, $b$ and $c$ "vary independently". Moreover, for any pair $(b, c) \in \mathcal{B}_\rho \times \mathcal{C}_\rho$, we observe

$$\sum_{\{i \in I_\rho \text{ if } (b_i, c_i) \approx (b, c)\}} r_i s_i^3 = c_\rho$$

   where $c_\rho$ is a constant independent of $i$.

   In general, though, we find $\{(b_i, c_i)\}_{i \in I_\rho} = \bigcup_q \mathcal{B}_{\rho,q} \times \mathcal{C}_{\rho,q}$ for some partitions $\{\mathcal{B}_{\rho,q}\}_q$ and $\{\mathcal{C}_{\rho,q}\}_q$ of $\mathcal{B}_\rho$ and $\mathcal{C}_\rho$. In other words, $b$ and $c$ vary independently within each partition. Moreover, we observe that for any $(b, c) \in \mathcal{B}_{\rho,q} \times \mathcal{C}_{\rho,q}$

$$\sum_{\{i \in I_{\rho,q} \text{ if } (b_i, c_i) \approx (b, c)\}} r_i s_i^3 = c_{\rho,q}$$

   This split into partitions is a technical detail (the partitions are easy to find in practice) and the reader may wish to ignore it.

7. **Observe $\mathcal{B}_{\rho,q} = -\mathcal{C}_{\rho,q}$.** We check that for each $b \in \mathcal{B}$, there exists $c \in \mathcal{C}$ with $\langle b, c \rangle \approx -1$ (and vice versa with $\mathcal{B}$ and $\mathcal{C}$ swapped).

8. **Observe that $\mathcal{B}_{\rho,q}$ is approximately a $\rho$-set.** We check that for all $x \in G$ and $b \in \mathcal{B}$, there exists $b' \in \mathcal{B}$ such that $\rho(x) b \approx b'$ (that is, $\langle \rho(x) b, b' \rangle \approx 1$).

Putting these observations together, we arrive at our final expression for the contribution to the network due to a single $\mathcal{B}_{\rho,q}$ (whose indices we drop to simply call $\mathcal{B}$):

$$f_{\rho, \mathcal{B}}(z \mid x, y) = - \sum_{b, b' \in \mathcal{B}} b^\top \rho(z) b' \, \mathrm{ReLU}[b^\top \rho(x) a - a^\top \rho(y) b'].$$

This is precisely the Equation 4 that we stated when introducing the $\rho$-set circuit in Section 4.1.

---

[19]For exploratory work, we use standard $k$-means. For the interpretation string of the compact proof, we also try a modification of $k$-means that takes into account the symmetries due to the corresponding irrep $\rho$. We find that this modified $k$-means algorithm does not yield substantially different results from the original.

### B.5 SIGN IRREP AND MODULAR ARITHMETIC

Let us briefly extend our attention to groups with complex-valued irreps and consider cyclic groups $\mathbb{Z}/p\mathbb{Z}$, i.e. arithmetic modulo $p$. All irreps of cyclic groups are one-dimensional, looking like $\rho(x) = e^{2\pi i k x/p} = \cos(2\pi k x/p) + i\sin(2\pi k x/p)$. The modular arithmetic setting is studied by Yip et al. (2024), where it is found that models use an approximation trick involving the sum-of-cosines formula and integration over a single variable:

$$\int_{-\pi}^{\pi} \cos(z + 2\phi)\,\mathrm{ReLU}[\cos(x + \phi) + \cos(y + \phi)]d\phi$$

$$= \left|\cos\left(\frac{x - y}{2}\right)\right|\left|\frac{1}{2}\int_{-\pi}^{\pi}\cos(z + 2\phi)\left|\cos\left(\frac{x + y}{2} + \phi\right)\right|d\phi\right|$$

$$= \left|\cos\left(\frac{x - y}{2}\right)\right|\frac{2\cos(x + y - z)}{3}.$$

Here, analogously to the sign circuit of our setting, the sum of the group elements' embeddings is expressed as the embedding of their sum in order to compute the desired inequality with only a single sum/integral, instead of a double sum. (In this case, the extraneous $|\cos((x - y)/2)|$ is not removed; indeed, this additional term is the "Achilles' heel" of this strategy, and necessitates that the model use multiple irreps, even though each one is faithful (Zhong et al., 2024).)

### B.6 CONSTANT PROJECTION VECTORS

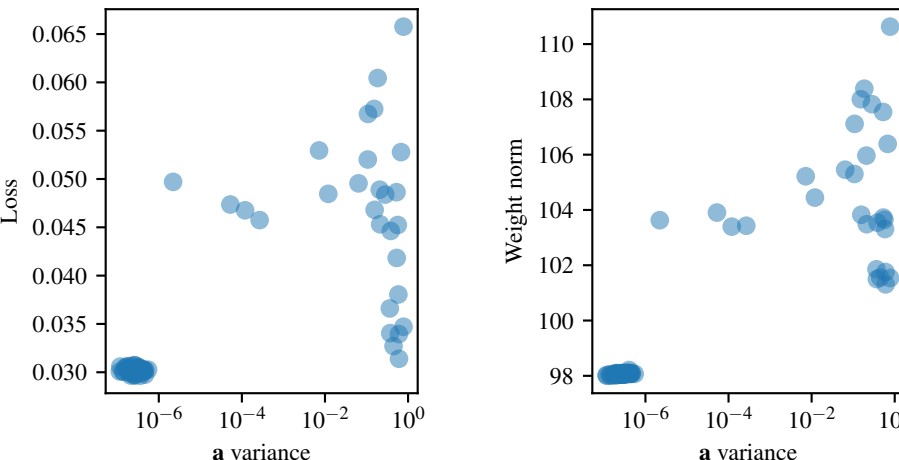

Figure 4: Plots of normalized variance $\mathbb{E}_i[\|\boldsymbol{a}_i - \mathbb{E}_i\boldsymbol{a}_i\|_2^2]/\mathbb{E}_i[\|\boldsymbol{a}_i\|_2^2]$ vs. model loss and weight norm, where $\boldsymbol{a}_i$ is the projection vector for neuron $i$, and expectation is taken across neurons within the 4d standard irrep of $S_5$. Each point is one model out of 100 trained on $S_5$. Notice that constant $\boldsymbol{a}_i$ across neurons is correlated with better model performance and lower weight norm.

Why does the network learn to set the $\boldsymbol{a}$ vector constant across neurons? A heuristic explanation is that such a constant vector $\boldsymbol{a}$ is one way to enforce bi-equivariance, which then leads the margin attained to be uniform across inputs $x, y \in G$. Morwani et al. (2024) show that this uniform margin must necessarily be the case at a maximum margin solution; further, models trained on cross-entropy loss in the zero weight decay limit indeed attain the maximum margin (Wei et al., 2019).

However, we do see that some fraction of trained models do not have constant $\boldsymbol{a}$; we do not have a full understanding of these models, and thus are unable to non-vacuously bound accuracy. We notice that these models tend to have inferior performance and higher weight norm, suggesting that they have converged to a poor local minimum by chance; see Figure 4. Further, even in these cases, we find that the $\boldsymbol{a}_i$ all lie in a two-dimensional subspace.

## C   Additional evidence for $\rho$-set circuits

In this section we provide additional evidence that models implement $\rho$-set circuits (Eq. 2). For each trained model, we constructed an idealized version of the model that implements $\rho$-set circuits exactly; each irrep $\rho$, corresponding $\rho$-set $\mathcal{B}$, and constant vector $\boldsymbol{a}$ are found automatically using the steps described in Section B.4.

Figure 5 plots the distance between original model parameters and parameters of idealized models. Figure 7 illustrates the effect of replacing each parameter type ($\boldsymbol{w}_l, \boldsymbol{w}_r, \boldsymbol{w}_u, \boldsymbol{w}_b$) in the original model with is idealized version. Figure 8 is the same but instead aggregated by neurons corresponding to each irrep. Figure 6 depicts the bi-equivariance of idealized models, original trained models, and randomly initialized models. We measure bi-equivariance by

$$\text{equiv}(\boldsymbol{\theta}) = \mathbb{E}_{z \in G} \left[ \frac{\sqrt{\text{Var}_{xy=z} f_{\boldsymbol{\theta}}(z \mid x, y)}}{\mathbb{E}_{xy=z} f_{\boldsymbol{\theta}}(z \mid x, y)} \right]. \tag{11}$$

If $f_{\boldsymbol{\theta}}$ is exactly bi-equivariant, then $\text{equiv}(\boldsymbol{\theta}) = 0$.

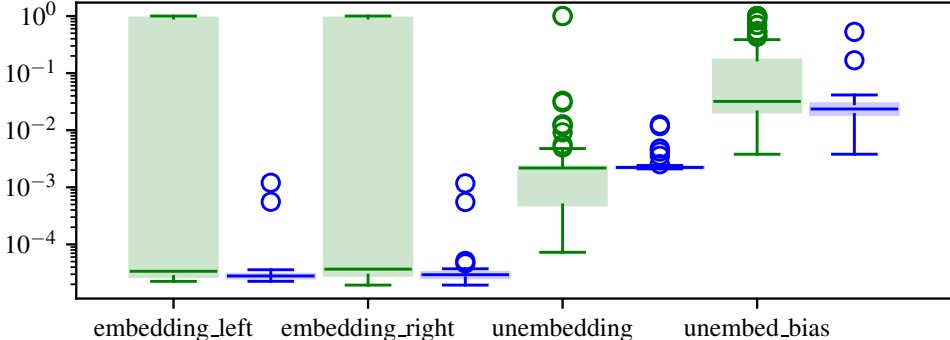

Figure 5: Normalized distance between original and idealized model parameters $\|\boldsymbol{w} - \hat{\boldsymbol{w}}\|_2^2 / \|\boldsymbol{w}\|_2^2$ (i.e. $1 - R^2$) for each of left embedding $\boldsymbol{w}_l$, right embedding $\boldsymbol{w}_r$, unembedding $\boldsymbol{w}_u$, and unembed bias $\boldsymbol{w}_b$ of 100 models trained on $S_5$. **Green** boxes include all models while **blue** boxes exclude models for which we find that the $\rho$-set explanation does not hold (i.e. either ($\boldsymbol{a}$-bad) or ($\rho$-bad)).

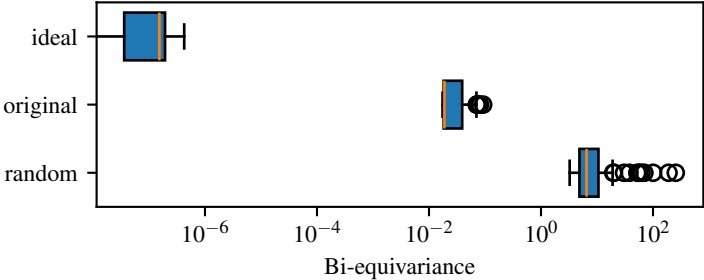

Figure 6: Bi-equivariance of idealized models, original trained models, and randomly initialized models for 100 models on $S_5$. See Eq. 11 for the definition of our bi-equivariance metric $\text{equiv}(\boldsymbol{\theta})$. Lower means more bi-equivariant and a value of zero means exactly bi-equivariant. Theoretically, the idealized model is exactly bi-equivariant when parameters are considered over $\mathbb{R}$; however, some non-bi-equivariance is introduced by floating point imprecision.

## D   Accuracy bounds via coset concentration

We construct a verifier $V_{\text{coset}}$ that takes as input $\boldsymbol{\theta}$ and an interpretation string

$$\pi = ((H_i, g_i))_{i=1}^m,$$

and returns a lower bound on $\alpha_G(\boldsymbol{\theta})$ in time $O(m|G|^2)$.

For each $x, y \in G$, the verifier $V_{\text{coset}}$ computes a lower bound on the margin

$$M(z) \leq \min_{\substack{x,y,z' \in G \\ xy=z \neq z'}} f_{\boldsymbol{\theta}}(xy \mid x, y) - f_{\boldsymbol{\theta}}(z' \mid x, y)$$

by doing the following:

1. Check that the $(H_i)_{i=1}^m$ are each subgroups of $G$ and construct the conjugate subgroups $K_i = g_i H_i g_i^{-1}$ in time $O(\sum_{i=1}^m |H_i|^2)$.

2. Construct idealized parameters $\tilde{\boldsymbol{\theta}}$ by averaging over the cosets given in $\pi$. Explicitly,

$$\tilde{\boldsymbol{w}}_l^i(x) = |H_i|^{-1} \sum_{x' \in H_i x} \boldsymbol{w}_l^i(x')$$

$$\tilde{\boldsymbol{w}}_r^i(y) = |K_i|^{-1} \sum_{y' \in yK_i} \boldsymbol{w}_r^i(y')$$

$$\tilde{\boldsymbol{w}}_u^i(z) = \begin{cases} |G - H_i g_i^{-1}|^{-1} \sum_{z' \notin H_i g_i^{-1}} \boldsymbol{w}_u^i(z') & z \notin H_i g_i \\ \min\{\boldsymbol{w}_u^i(z), \min_{z' \notin H_i g_i} \tilde{\boldsymbol{w}}_u^i(z')\} & z \in H_i g_i \end{cases}$$

$$\tilde{\boldsymbol{w}}_b = \mathbf{0}.$$

This takes time $O(|G|m)$.

3. Compute $s(H_i, g_i)$ for each $i$ according to Eq 8. This takes time $O(\sum_{i=1}^m |H_i \backslash G|^2)$.

4. For each $z \in G$, compute

$$M(z) = \min_{\substack{z' \in G \\ z' \neq z}} \sum_{(H,g)} s(H,g)\mathbf{1}\{z \notin Hg^{-1}\} \sum_{\substack{i \in [m] \\ (H_i, g_i) = (H,g)}} (\boldsymbol{w}_u^i(z) - \boldsymbol{w}_u^i(z'))$$

This takes time $O(m|G|^2)$.

5. Lemma G.2 gives an upper bound on the $\ell_\infty$ norm between the output logits of original and idealized models for each $x, y \in G$ in time $O(m|G|^2)$. Sum the difference in logit values of the original and idealized models on the correct answer, which again can be computed in time $O(m|G|^2)$. This gives

$$L(x,y) \geq \|f_{\boldsymbol{\theta}}(\cdot \mid x, y) - f_{\tilde{\boldsymbol{\theta}}}(\cdot \mid x, y)\|_\infty + f_{\tilde{\boldsymbol{\theta}}}(xy \mid x, y) - f_{\boldsymbol{\theta}}(xy \mid x, y). \tag{12}$$

The final accuracy bound is

$$\Pr_{z \sim \text{Unif}(G)}\left[M(z) > \max_{\substack{x,y \in G \\ xy=z}} L(x,y)\right].$$

The total time complexity is dominated by the last two steps. Soundness, i.e. $\forall \pi : V_{\text{coset}}(\boldsymbol{\theta}, G, \pi) \leq \alpha_G(\boldsymbol{\theta})$, follows from Proposition A.2.

**The bias term** The description of the coset circuit in Stander et al. (2024) makes no mention of the unembedding bias term, although it is present in the models they train for their experiments. In practice, we find that the bias term is large. The maximum minus minimum value of the bias would then be added directly to $L(x,y)$ for every $x, y \in G$ if we were to run $V_{\text{coset}}$ as written. Thus, in addition, we train models without an explicit bias term for our $V_{\text{coset}}$ experiments. We find that such models are qualitatively similar to models with an explicit bias; the missing bias term is simply added uniformly to each unembedding weight $\boldsymbol{w}_u^i$. Perhaps because of this, we are still unable to obtain nonvacuous bounds from $V_{\text{coset}}$ even for these bias-less models.

## E   ACCURACY BOUNDS VIA $\rho$-SETS

We construct a verifier $V_{\text{irrep}}$ that takes as input $\boldsymbol{\theta}$ and an interpretation string $\pi$ and returns a lower bound on $\alpha_G(\boldsymbol{\theta})$ in time $O(m|G|^2)$. The interpretation $\pi$ comprises $((\rho_i, q_i, \boldsymbol{a}_i, \boldsymbol{b}_i, c_i)_{i=1}^m)$, where $q_i$ is the index of the corresponding $\rho$-set $\mathcal{B}_{\rho,q} = \{\boldsymbol{b}_i \mid \rho_i = \rho, q_i = q\}$. The verifier then does

1. Check that each $\mathcal{B}_{\rho,q}$ is indeed a $\rho$-set. This takes time $O(\sum_{\rho,q}|G||\mathcal{B}_{\rho,q}|^2\dim(\rho)^2)$. Since $m \geq \sum_{\rho,q}|\mathcal{B}_{\rho,q}|^2$ and $\sum_{\rho\in\text{Irrep}(G)}\dim(\rho)^2 = |G|$, this is upper-bounded by $O(|G|^2 m)$.

2. Within neurons corresponding to each $(\rho, q)$, check that $\boldsymbol{a}_i$ is constant. This again takes time no more than $O(|G|m)$.

3. For each $(\rho, q)$ where $\rho$ is not the sign irrep, check that the coefficients $c_i$ across all neurons corresponding to $(\rho, q)$ is constant. This takes time $O(m)$.

4. For neurons corresponding to the sign irrep, there is only one $\rho$-set $\mathcal{B}_\rho = \{\pm 1\}$. Check that the constraint in Observation B.2(8) holds; that is, that the neurons corresponding to $(+1, +1)$ have coefficients summing to the same value as those corresponding to $(-1, -1)$, and likewise for $(+1, -1)$ and $(-1, +1)$.

5. Construct the idealized parameters $\tilde{\boldsymbol{\theta}}$ consisting of

$$\tilde{\boldsymbol{w}}_l^i(x) = \boldsymbol{b}_i^\top \rho(x)\boldsymbol{a}_i$$
$$\tilde{\boldsymbol{w}}_r^i(y) = \boldsymbol{a}_i^\top \rho(x)\boldsymbol{c}_i$$
$$\tilde{\boldsymbol{w}}_u^i(z) = c_i \boldsymbol{b}_i^\top \rho(z)\boldsymbol{b}_j$$
$$\tilde{\boldsymbol{w}}_b(z) = (c_- - c_+)\,\text{sgn}(z).$$

This takes time $O(\sum_{i=1}^m \dim(\rho_i)^2 |G|) \leq O(|G|^2 m)$.

6. Compute the idealized margin

$$M = \min_{x,y\in G} f_{\tilde{\boldsymbol{\theta}}}(xy \mid x, y) - \max_{z'\neq xy} f_{\tilde{\boldsymbol{\theta}}}(z \mid xy).$$

By Proposition B.3, this can be done with a single forward pass of $f_{\tilde{\boldsymbol{\theta}}}$, in time $O(|G|m)$.

7. Use Lemma G.2 to compute $L$ as in Eq 12 in time $O(m|G|^2)$. The final accuracy bound is

$$\Pr_{x,y\sim\text{Unif}(G)}[M > L(x,y)].$$

The total time complexity is dominated by the last step. Soundness, i.e. $\forall \pi : V_{\text{irrep}}(\boldsymbol{\theta}, G, \pi) \leq \alpha_G(\boldsymbol{\theta})$, is obvious from construction and Proposition B.3.

## F  INSUFFICIENCY OF CAUSAL INTERVENTIONS

Stander et al. (2024) perform a series of causal interventions on a model trained on $S_5$, and find results consistent with their description of the cosets algorithm. However, we expect that these interventions would have the same result for models implementing different algorithms. To verify this, we replicate their outcomes with a model trained on the cyclic group $G = \mathbb{Z}/53\mathbb{Z}$; such a model cannot be using the coset algorithm, as $G$ as no non-trivial subgroups. Models trained cyclic groups were studied in Yip et al. (2024) and found to be using a distinct algorithm; see also discussion in Section B.5.

In detail, the interventions that Stander et al. (2024) perform on $S_5$ are:

1. **Embedding exchange:** Swapping the model's left and right embeddings destroys model performance. Since $S_5$ is non-commutative, we expect this to be the case regardless of what algorithm the model is implementing. Even with $\mathbb{Z}/53\mathbb{Z}$, which is commutative, we get this result, since $\boldsymbol{W}_l\boldsymbol{E}_l(x) + \boldsymbol{W}_r\boldsymbol{E}_r(y) \neq \boldsymbol{W}_l\boldsymbol{E}_r(y) + \boldsymbol{W}_r\boldsymbol{E}_l(x)$.

2. **Switch permutation sign** Multiplying either the left or right embeddings individually by $-1$ destroys model performance, while multiplying both preserves model performance. We find this to be the case with $\mathbb{Z}/53\mathbb{Z}$ as well.

3. **Absolute value non-linearity** Replacing the ReLU nonlinearity with an absolute value improves model performance. Again, this is the case with $\mathbb{Z}/53\mathbb{Z}$ as well. We explain this by decomposing $\text{ReLU}(x) = (x + |x|)/2$. By inspecting the summation in Eq 2, we see that the $x/2$ component sums to zero, so only the $|x|/2$ term contributes. Thus, replacing the ReLU with an absolute value is approximately equivalent to doubling the activations, which reduces loss assuming the model already has near-perfect accuracy.

4. **Distribution change** Perturbing model activations by $\mathcal{N}(1,1)$ reduces performance to a greater extent than perturbing with $\mathcal{N}(1,-1)$. Again, we observe this with $\mathbb{Z}/53\mathbb{Z}$.

See Table 1 for results.

Table 1: Causal interventions aggregated over 128 runs on $\mathbb{Z}/53\mathbb{Z}$ (ours) juxtaposed with the same interventions aggregated over 128 runs on $S_5$ (Stander et al., 2024). We train our models with fewer iterations than Stander et al. (2024), resulting in higher base loss. However, the directional effect of each intervention is the same, even though the coset concentration explanation does not hold for $\mathbb{Z}/53\mathbb{Z}$.

| | $\mathbb{Z}/53\mathbb{Z}$ (ours) | | $S_5$ (Stander et al., 2024) | |
|---|---|---|---|---|
| **Intervention** | **Mean accuracy** | **Mean loss** | **Mean accuracy** | **Mean loss** |
| Base model | 99.55% | 0.0711 | 99.99% | 1.97e-6 |
| Embedding swap | 03.85% | 4.15 | 1% | 4.76 |
| Switch left and right sign | 99.69% | 0.0663 | 100% | 1.97e-6 |
| Switch left sign | 00.00% | 17.2 | 0% | 22.39 |
| Switch right sign | 00.00% | 17.2 | 0% | 22.36 |
| Absolute value nonlinearity | 99.88% | 0.0045 | 100% | 3.69e-13 |
| Perturb $\mathcal{N}(0,1)$ | 76.90% | 0.829 | 97.8% | 0.0017 |
| Perturb $\mathcal{N}(0,.1)$ | 99.51% | 0.0752 | 99.99% | 2.96e-6 |
| Perturb $\mathcal{N}(1,1)$ | 49.80% | 1.79 | 88% | 0.029 |
| Perturb $\mathcal{N}(1,-1)$ | 83.17% | 0.780 | 98% | 0.0021 |

# G   ADDITIONAL PROOFS

**Lemma G.1.** *Let*

$$\boldsymbol{\theta} = (\boldsymbol{U}, \boldsymbol{E}_l, \boldsymbol{E}_r, \boldsymbol{w}_b) = ((\boldsymbol{w}_u^i, \boldsymbol{w}_l^i, \boldsymbol{w}_r^i)_{i=1}^m, \boldsymbol{w}_b),$$
$$\tilde{\boldsymbol{\theta}} = (\tilde{\boldsymbol{U}}, \tilde{\boldsymbol{E}}_l, \tilde{\boldsymbol{E}}_r, \tilde{\boldsymbol{w}}_b) = ((\tilde{\boldsymbol{w}}_u^i, \tilde{\boldsymbol{w}}_l^i, \tilde{\boldsymbol{w}}_r^i)_{i=1}^m, \tilde{\boldsymbol{w}}_b)$$

*Then, for any $x, y \in G$,*

$$\max_{z \in G} |f_{\boldsymbol{\theta}}(z \mid x, y) - f_{\boldsymbol{\theta}'}(z \mid x, y)|$$

$$= \max_{z \in G} \left| \sum_{i=1}^{m} \left( \boldsymbol{w}_u^i(z) \operatorname{ReLU}[\boldsymbol{w}_l^i(x) + \boldsymbol{w}_l^i(y)] + \boldsymbol{w}_b(z) - \tilde{\boldsymbol{w}}_u(z) \operatorname{ReLU}[\tilde{\boldsymbol{w}}_l(x) + \tilde{\boldsymbol{w}}_l(y)] - \tilde{\boldsymbol{w}}_b(z) \right) \right|$$

$$\leq \max_{z \in G} \left| \sum_{i=1}^{m} (\boldsymbol{w}_u^i(z) - \tilde{\boldsymbol{w}}_u^i(z)) \operatorname{ReLU}[\boldsymbol{w}_l^i(x) + \boldsymbol{w}_l^i(y)] \right|$$

$$+ \max_{z \in G} \left| \sum_{i=1}^{m} \tilde{\boldsymbol{w}}_u^i(z) \left( \operatorname{ReLU}[\boldsymbol{w}_l^i(x) + \boldsymbol{w}_l^i(y)] - \operatorname{ReLU}[\tilde{\boldsymbol{w}}_l^i(x) + \tilde{\boldsymbol{w}}_l^i(y)] \right) \right|$$

$$+ \max_{z \in G} |\boldsymbol{w}_b(z) - \tilde{\boldsymbol{w}}_b(z)|$$

$$= \|(\boldsymbol{U} - \tilde{\boldsymbol{U}})^\top \operatorname{ReLU}[\boldsymbol{E}_l(x) + \boldsymbol{E}_r(y)]\|_\infty$$

$$+ \left\| \tilde{\boldsymbol{U}}^\top \left( \operatorname{ReLU}[\boldsymbol{E}_l(x) + \boldsymbol{E}_r(y)] - \operatorname{ReLU}[\tilde{\boldsymbol{E}}_l(x) + \tilde{\boldsymbol{E}}_r(y)] \right) \right\|_\infty$$

$$+ \|\boldsymbol{w}_b - \tilde{\boldsymbol{w}}_b\|_\infty$$

$$\leq \|(\boldsymbol{U} - \tilde{\boldsymbol{U}})^\top\|_{2,\infty} \|\operatorname{ReLU}[\boldsymbol{E}_l(x) + \boldsymbol{E}_r(y)]\|_2$$

$$+ \|\tilde{\boldsymbol{U}}^\top\|_{2,\infty} \| \left( \operatorname{ReLU}[\boldsymbol{E}_l(x) + \boldsymbol{E}_r(y)] - \operatorname{ReLU}[\tilde{\boldsymbol{E}}_l(x) + \tilde{\boldsymbol{E}}_r(y)] \right) \|_2$$

$$+ \|\boldsymbol{w}_b - \tilde{\boldsymbol{w}}_b\|_\infty$$

$$\leq \|(\boldsymbol{U} - \tilde{\boldsymbol{U}})^\top\|_{2,\infty} (\|\boldsymbol{E}_l(x)\|_{1,2} + \|\boldsymbol{E}_r(y)\|_{1,2})$$

$$+ \|\tilde{\boldsymbol{U}}^\top\|_{2,\infty} (\|\boldsymbol{E}_l - \tilde{\boldsymbol{E}}_l\|_{1,2} + \|\boldsymbol{E}_r - \tilde{\boldsymbol{E}}_r\|_{1,2})$$

$$+ \|\boldsymbol{w}_b - \tilde{\boldsymbol{w}}_b\|_\infty.$$

$\square$

**Lemma G.2.** *Let $\boldsymbol{\theta}$ and $\tilde{\boldsymbol{\theta}}$ be as in Lemma G.1. Further, suppose we have a margin lower bound function for $\boldsymbol{\theta}$*

$$M(z) \leq \min_{\substack{x,y,z' \in G \\ xy = z \neq z'}} f_{\boldsymbol{\theta}}(xy \mid x, y) - f_{\boldsymbol{\theta}}(z' \mid x, y).$$

*Then,*

$$\alpha_{\mathrm{G}}(\tilde{\boldsymbol{\theta}}) \geq \Pr_{x,y \sim \mathrm{Unif}(G)} \left[ M(xy) > \|(\boldsymbol{U} - \tilde{\boldsymbol{U}})^\top\|_{2,\infty} \|\operatorname{ReLU}[\boldsymbol{E}_l(x) + \boldsymbol{E}_r(y)]\|_2 \right. \tag{13}$$

$$+ \|\tilde{\boldsymbol{U}}^\top\|_{2,\infty} \left\| \left( \operatorname{ReLU}[\boldsymbol{E}_l(x) + \boldsymbol{E}_r(y)] - \operatorname{ReLU}[\tilde{\boldsymbol{E}}_l(x) + \tilde{\boldsymbol{E}}_r(y)] \right) \right\|_2$$

$$\left. + \|\boldsymbol{w}_b - \tilde{\boldsymbol{w}}_b\|_\infty \right]$$

$$\geq \Pr_{z \sim \mathrm{Unif}(G)} \left[ M(z) > \|(\boldsymbol{U} - \tilde{\boldsymbol{U}})^\top\|_{2,\infty} (\|\boldsymbol{E}_l(x)\|_{1,2} + \|\boldsymbol{E}_r(y)\|_{1,2}) \right. \tag{14}$$

$$+ \|\tilde{\boldsymbol{U}}^\top\|_{2,\infty} (\|\boldsymbol{E}_l - \tilde{\boldsymbol{E}}_l\|_{1,2} + \|\boldsymbol{E}_r - \tilde{\boldsymbol{E}}_r\|_{1,2})$$

$$\left. + \|\boldsymbol{w}_b - \tilde{\boldsymbol{w}}_b\|_\infty \right].$$

*The bound in Equation 13 can be computed in time $O(m|G|^2)$, while that in Equation 14 can be computed in time $O(m|G|)$.*

*Proof.* This follows immediately from Lemma G.1. $\square$

For the remainder of this section, let $\ell$ denote the cross-entropy loss:

$$\ell(\boldsymbol{x}, i) := -\log \frac{e^{\boldsymbol{x}_i}}{\sum_j \boldsymbol{x}_j}.$$

**Lemma G.3.** *The cross-entropy loss is $\sqrt{2}$-Lipschitz.*

*Proof.* Cross-entropy loss $\ell$ is differentiable with

$$\nabla_{\boldsymbol{x}}(\ell(\boldsymbol{x}, i))_j = \frac{e^{\boldsymbol{x}_j} - \delta_{ij} \sum_k e^{\boldsymbol{x}_k}}{\sum_k e^{\boldsymbol{x}_k}}.$$

Hence,

$$\|\nabla_{\boldsymbol{x}}\ell(\boldsymbol{x}, i)\|_2^2 = \frac{(\sum_{k \neq i} e^{\boldsymbol{x}_k})^2 + \sum_{k \neq i} e^{2\boldsymbol{x}_k}}{(\sum_k e^{\boldsymbol{x}_k})^2} \leq 2.$$

$\square$

**Lemma G.4.** *For $\boldsymbol{x}, \boldsymbol{y} \in \mathbb{R}^n$ and $i \in [n]$, the cross-entropy loss $\ell(\cdot, i)$ satisfies*

$$\ell(\boldsymbol{y}, i) \leq \ell(\boldsymbol{x}, i) + \nabla\ell(\boldsymbol{x})^\top(\boldsymbol{y} - \boldsymbol{x}) + \frac{1}{4}\|\boldsymbol{x} - \boldsymbol{y}\|_2^2$$

$$\leq \ell(\boldsymbol{x}, i) + \|\nabla\ell(\boldsymbol{x})\|_2\|\boldsymbol{x} - \boldsymbol{y}\|_2 + \frac{1}{4}\|\boldsymbol{x} - \boldsymbol{y}\|_2^2.$$

*Proof.* It suffices to show $0 \preceq \nabla^2\ell(\boldsymbol{x}, i) \preceq \frac{1}{2}\boldsymbol{I}$, whence $\ell(\cdot, i)$ is convex and $1/2$-smooth; the desired inequality is then a well-known consequence (Beck, 2017).

Write $\boldsymbol{p}_i = \frac{e^{\boldsymbol{x}_i}}{\sum_{j=1}^n e^{\boldsymbol{x}_j}}$. Then (Boyd & Vandenberghe, 2004),

$$\nabla^2\ell(\boldsymbol{x}, i) = \nabla^2 \log\left(\sum_{j=1}^n e^{\boldsymbol{x}_j}\right) = \mathrm{diag}(\boldsymbol{p}) - \boldsymbol{p}\boldsymbol{p}^\top,$$

so, for any vector $\boldsymbol{v} \in \mathbb{R}^n$,

$$\boldsymbol{v}^\top\nabla^2\ell_i(\boldsymbol{x})\boldsymbol{v} = \boldsymbol{v}^\top(\mathrm{diag}(\boldsymbol{p}) - \boldsymbol{p}\boldsymbol{p}^\top)\boldsymbol{v} = \mathrm{Var}_{\boldsymbol{p}}(\boldsymbol{v}) \geq 0,$$

confirming that $\nabla^2\ell(\boldsymbol{x}, i) \succeq 0$. Furthermore, applying the Gershgorin circle theorem to the Hessian,

$$\lambda_{\max}(\nabla^2\ell(\boldsymbol{x}, i)) \leq \max_{j \in [n]}\left(\boldsymbol{p}_j - \boldsymbol{p}_j^2 + \sum_{k \neq j} \boldsymbol{p}_j\boldsymbol{p}_k\right) = \max_{j \in [n]} 2\boldsymbol{p}_j(1 - \boldsymbol{p}_j) \leq \frac{1}{2},$$

so $\nabla^2\ell(\boldsymbol{x}, i) \preceq \frac{1}{2}\boldsymbol{I}$. $\square$

**Lemma G.5.** *Let $\rho\colon G \to \mathrm{GL}(n, \mathbb{R})$ be a permutation representation of $G$ that decomposes into two subspaces $V \oplus W$ such that $\rho$ acts trivially on $W$. (For example, the standard $(n-1)$-dimensional irrep of $S_n$ is of this form.) Let $\boldsymbol{V}$ and $\boldsymbol{W}$ be orthonormal bases of $V, W \in \mathbb{R}^n$, respectively, and let $\boldsymbol{P} = \boldsymbol{V}\boldsymbol{V}^\top$ and $\boldsymbol{Q} = \boldsymbol{W}\boldsymbol{W}^\top$ be the corresponding orthogonal projections, so that $\boldsymbol{P} + \boldsymbol{Q} = \boldsymbol{I}$. Denote $\boldsymbol{b}_i = \boldsymbol{V}^\top\boldsymbol{e}_i$, where $(\boldsymbol{e}_i)_{i=1}^n$ are the standard basis of $\mathbb{R}^n$. Then, for any $\boldsymbol{a} \in V$,*

$$\varphi(z) = \left\langle \rho|_V(z), -\sum_{i,j}^n \mathrm{ReLU}[\boldsymbol{a}^\top(\boldsymbol{b}_i - \boldsymbol{b}_j)]\boldsymbol{b}_j\boldsymbol{b}_i^\top \right\rangle$$

*is maximized at $z = e$.*

*Proof.*

$$\varphi(z) = -\left\langle \rho|_V(z), \sum_{i,j}^n \mathrm{ReLU}[\boldsymbol{a}^\top(\boldsymbol{b}_i - \boldsymbol{b}_j)]\boldsymbol{b}_j\boldsymbol{b}_i^\top \right\rangle$$

$$= -\left\langle \rho|_V(z), \sum_{i,j}^n \mathrm{ReLU}[\boldsymbol{a}^\top(\boldsymbol{b}_i - \boldsymbol{b}_j)]\boldsymbol{V}^\top\boldsymbol{e}_j\boldsymbol{e}_i^\top\boldsymbol{V} \right\rangle$$

$$= -\left\langle \boldsymbol{V}\rho|_V(z)\boldsymbol{V}^\top, \sum_{i,j}^n \mathrm{ReLU}[\boldsymbol{a}^\top(\boldsymbol{b}_i - \boldsymbol{b}_j)]\boldsymbol{e}_j\boldsymbol{e}_i^\top \right\rangle$$

$$= -\left\langle \boldsymbol{P}\rho(z)\boldsymbol{P}, \sum_{i,j}^n \mathrm{ReLU}[\boldsymbol{a}^\top(\boldsymbol{b}_i - \boldsymbol{b}_j)]\boldsymbol{e}_j\boldsymbol{e}_i^\top \right\rangle$$

$$= -\left\langle \rho(z), \sum_{i,j}^n \mathrm{ReLU}[\boldsymbol{a}^\top(\boldsymbol{b}_i - \boldsymbol{b}_j)]\boldsymbol{e}_j\boldsymbol{e}_i^\top \right\rangle$$

$$\quad - \left\langle \boldsymbol{Q}\rho(z)\boldsymbol{Q}^\top, \sum_{i,j}^n \mathrm{ReLU}[\boldsymbol{a}^\top(\boldsymbol{b}_i - \boldsymbol{b}_j)]\boldsymbol{e}_j\boldsymbol{e}_i^\top \right\rangle$$

$$= -\left\langle \rho(z), \sum_{i,j}^n \mathrm{ReLU}[\boldsymbol{a}^\top(\boldsymbol{b}_i - \boldsymbol{b}_j)]\boldsymbol{e}_j\boldsymbol{e}_i^\top \right\rangle$$

$$\quad - \left\langle \boldsymbol{Q}\boldsymbol{Q}^\top, \sum_{i,j}^n \mathrm{ReLU}[\boldsymbol{a}^\top(\boldsymbol{b}_i - \boldsymbol{b}_j)]\boldsymbol{e}_j\boldsymbol{e}_i^\top \right\rangle,$$

where the last step uses that $\rho|_W$ is trivial. The first term of the last line is the negation of a sum over off-diagonal entries of the permutation matrix $\rho(z)$, and thus maximized at $z = e$. The second term does not depend on $z$. $\qquad\square$

An irrep $V$ of $G$ admits such a $\rho$ with $W$ being the trivial representation, if for the corresponding subgroup $H \subset G$ the double coset has two elements $H \setminus G/H$. This applies in our case where $G = S_5$ and $H = S_4$.

**Lemma G.6.** *Suppose $f\colon G \to \mathbb{C}$ is* coset concentrated *and* irrep sparse*; that is, $f$ is constant on the cosets of some $H \le G$ and there exists $\rho \in \mathrm{Irrep}(G)$ mapping $\rho\colon G \to \mathrm{GL}(d, \mathbb{C})$ such that $f$ is a linear combination of the entries of $\rho$. Then, there must exist an embedding $\iota\colon G/H \to \mathbb{C}^d$ such that the action of $\rho$ on $\iota(G/H)$ is isomorphic to the action of $G$ on $G/H$. Further, there exist $\boldsymbol{a} \in (\mathbb{C}^d)^*$ and $\boldsymbol{b} \in \mathbb{C}^d$ such that $f(g) = \langle \boldsymbol{a}, \rho(g)\boldsymbol{b}\rangle$.*

*The converse also holds and is immediate.*

*Proof.* Let $\{g_1, \ldots, g_k\}$ be representatives for the cosets $G/H$ with $g_1 = e$, and let $\tilde{\rho}\colon G \to S_k \subseteq \mathrm{GL}(\bigoplus_{i=1}^k g_i\mathbb{C}) =: V$ be the permutation representation corresponding to the action of $G$ on $G/H$. (That is, $\tilde{\rho}$ is induced by the trivial representation on $H$.) Let $\{\boldsymbol{e}_{g_i}\}_{i=1}^k$ denote the basis vectors of $V$. Define $\boldsymbol{a} \in V^*$ by $\langle \boldsymbol{a}, \boldsymbol{e}_{g_i}\rangle = f(g_i)$ for each $i \in [k]$. Hence $f(g_i) = \langle \boldsymbol{a}, \tilde{\rho}(g_i)\boldsymbol{e}_{g_1}\rangle$. Since $f$ is coset concentrated, this same relation holds for all $g \in G$. Now, let $V = \bigoplus_{j=1}^n V_j$ be the decomposition of $V$ into irreps, with corresponding projections $\pi_j\colon V \to V_j$ and irreps $\rho_j\colon G \to \mathrm{GL}(V_j)$. We have

$$f(g) = \langle \boldsymbol{a}, \tilde{\rho}(g)\boldsymbol{e}_{g_1}\rangle = \sum_{j=1}^n \langle \boldsymbol{a}|_{V_j}, \pi_j\tilde{\rho}(g)\boldsymbol{e}_{g_1}\rangle = \sum_{j=1}^n \langle \boldsymbol{a}|_{V_j}, \rho_j(g)\pi_j\boldsymbol{e}_{g_1}\rangle.$$

By the irrep sparsity assumption, at most one term of this sum is nonzero, and that corresponding term must have $\rho_j = \rho$:

$$f(g) = \langle \boldsymbol{a}|_{V_j}, \rho(g)\pi_j\boldsymbol{e}_{g_1}\rangle.$$

Then $\rho$ acts on $\{\pi_j\boldsymbol{e}_{g_i}\}_{i=1}^k$ isomorphically to the action of $G$ on $G/H$ and $(\boldsymbol{a}|_{V_j}, \pi_j\boldsymbol{e}_{g_1})$ is the desired covector and vector pair from the theorem statement. $\qquad\square$

## H  PERMUTATION REPRESENTATIONS AND $\rho$-SETS OF $S_5$

In this section we enumerate all irreps of $S_5$ and their corresponding minimum $\rho$-sets.

| Irrep | Minimum $\rho$-set size | Stabilizer |
|---|---|---|
| trivial (`1d-0`) | 1 | $S_5$ |
| sign (`1d-1`) | 2 | $A_5$ |
| standard (`4d-0`) | 5 | $S_4$ |
| sign-standard (`4d-1`) | 10 | $A_4$ |
| `5d-0` | 6 | $F_5$ |
| `5d-1` | 12 | $D_{12}$ |
| `6d-0` | 20 | $\mathbb{Z}/6\mathbb{Z}$ or $S_3$ |

Table 2: Irreps $\rho \in \mathrm{Irrep}(S_5)$ by the size of the minimum $\rho$-set, and corresponding stabilizers. We name each irrep by its dimension and an arbitrary disambiguating integer; e.g. `5d-0` is a five-dimensional irrep. A projected $\rho$-set (Definition 6.3) is constant on the cosets of its stabilizer. Notice that the ordering of $\mathrm{Irrep}(S_5)$ by minimum $\rho$-set size matches the ordering by frequencies with which irreps are learned (Chughtai et al., 2023, Figure 7).

| Stabilizer | $G$-set size | Irreps present |
|---|---|---|
| $S_5$ | 1 | `1d-0` |
| $A_5$ | 2 | `1d-0`,`1d-1` |
| $S_4$ | 5 | `1d-0`,`4d-0` |
| $F_5$ | 6 | `1d-0`,`5d-0` |
| $A_4$ | 10 | `1d-0`,`1d-1`,`4d-0`,`4d-1` |
| $D_{12}$ | 10 | `1d-0`,`4d-0`,`5d-1` |
| $D_{10}$ | 12 | `1d-0`,`1d-1`,`5d-0`,`5d-1` |
| $D_8$ | 15 | `1d-0`,`4d-0`,`5d-0`,`5d-1` |
| $\mathbb{Z}/6\mathbb{Z}$ | 20 | `1d-0`,`4d-0`,`4d-1`,`5d-1`,`6d-0` |
| $S_3^0$ | 20 | `1d-0`,`4d-0`,`5d-1`,`6d-0` |
| $S_3^1$ | 20 | `1d-0`,`1d-1`,`4d-0`,`4d-1`,`5d-0`,`5d-1` |
| $\mathbb{Z}/5\mathbb{Z}$ | 24 | `1d-0`,`1d-1`,`5d-0`,`5d-1`,`6d-0` |
| $V_4^0$ | 30 | `1d-0`,`1d-1`,`4d-0`,`4d-1`,`5d-0`,`5d-1` |
| $V_4^1$ | 30 | `1d-0`,`4d-0`,`5d-0`,`5d-1`,`6d-0` |
| $\mathbb{Z}/4\mathbb{Z}$ | 30 | `1d-0`,`4d-0`,`4d-1`,`5d-0`,`5d-1`,`6d-0` |

Table 3: All transitive permutation representations of $S_5$ with size no more than 30 along with decomposition into irreps. By the orbit-stabilizer theorem, transitive permutation representations correspond directly to left actions of $S_5$ on cosets of its subgroups; the subgroup acted upon is then the stabilizer of the action. Upper indices disambiguate subgroups of $S_5$ that are isomorphic but not conjugate. $F_5$ is the Frobenius group of order 20 and $V_4$ is the Klein four-group.

## I  EXPERIMENT DETAILS

For the main text, we train 100 one-hidden-layer models with hidden dimensionality $m = 128$ on the group $S_5$. The test set is all pairs of two inputs from $S_5$, with $|S_5|^2 = 14400$ points total. The training set comprises iid samples from the test set and has size 40% of the test set. Note that we use the same training set for each of the 100 training runs. Each model was trained over 25000 epochs. Learning rate was set to 1e-2. We use the Adam optimizer (Kingma & Ba, 2015) with weight decay 2e-4 and $(\beta_1, \beta_2) = (0.9, 0.98)$.[20] All models were trained on one Nvidia A6000 GPU. Compact

---

[20]Note that previous work uses AdamW (Loshchilov & Hutter, 2019) instead, with weight decay 1. However, we found that models trained with Adam grok the group composition task in an order of magnitude fewer epochs. We did not notice significant differences in the end result post-grokking between models trained with Adam vs AdamW.

proof verifiers were run on an Intel Core i5-1350P CPU. Neural networks were implemented in PyTorch (Paszke et al., 2019). Their group-theoretic properties were analyzed with GAP (GAP, 2024).

In Section K.1, models are trained with the same hyperparameters as described above, except 1)

- For $A_5$, the hidden dimensionality is $m = 256$, the weight decay is $10^{-6}$, and the unembedding bias is omitted. Recall that, in our explanation, the role of the unembedding is to deal with the sign irrep, which is not present for alternating groups. The larger hidden dimensionality and smaller weight decay were used in an attempt to reduce occurrences of ($\rho$-bad), though we did not observe these changes to have significant effect
- For $S_4$, the hidden dimensionality $m = 64$ and the training set is 80% of the test set in order to account for the smaller total number of data points.

As input to the verifier $V_{\text{coset}}$, recall that the interpretation string looks like $\pi = ((H_i, g_i))_{i=1}^m$, where the left embedding at neuron $i \in [m]$ should be constant on the right cosets of $H_i$ and the right embedding at neuron $i$ should be constant on the left cosets of $g_i H_i g_i^{-1}$. When constructing $\pi$, we set each $H_i$ to be the largest subgroup of $G$ such that

$$\frac{\mathbb{E}\,\text{Var}(\boldsymbol{w}_l^i \mid H_i \backslash G)}{\text{Var}(\boldsymbol{w}_l^i)} < 0.01,$$

and $K_i$ to the largest subgroup such that

$$\frac{\mathbb{E}\,\text{Var}(\boldsymbol{w}_l^i \mid G/K_i)}{\text{Var}(\boldsymbol{w}_l^i)} < 0.01,$$

and check for the existence of $g_i$ such that $K_i = g_i H_i g_i^{-1}$. The quotient on the LHS is bounded in $[0, 1]$ by the law of total variance.

As input to the verifier $V_{\text{irrep}}$, the interpretation string is found using an automated version of the steps discussed in Section B.4. The automated process labels each neuron with an $\rho_i \in \text{Irrep}(G)$ and a corresponding $\rho_i$-set $\mathcal{B}$. The irrep $\rho_i$ is chosen to have the largest $R^2$ against $\boldsymbol{w}_l^i$, or none, if the $R^2$ if no irrep exceeds 95%. The $\rho$-set is recovered using singular value decomposition of the coefficient matrices $\boldsymbol{A}_i, \boldsymbol{B}_i, \boldsymbol{C}_i$ as defined in Section B.4 and a variant of $k$-means clustering. We find that the clustering step is the most fragile part of the interpretation creation process—in practice, for each model, we run the process several times and choose the interpretation string that yields the highest accuracy bounds from the verifier. Note also that the construction of the interpretation string $\pi$ does not count towards the runtime of the verifier (see Section 5).

## J  BOUNDING THE LOSS

In this paper we focus on lower-bounding the test-set accuracy. Another natural choice is the test-set cross-entropy loss, which contains information about the model's *confidence* in its answers that accuracy obscures. Note that, in principle, the compact proofs framework can be applied just as well to this metric, or indeed any well-defined quantity associated to the model. We view this flexibility as a strength of the framework. However, those using compact proofs to evaluate model interpretations must take care that the choice of quantity being bounded is relevant to the task.

We consider two simple techniques for reusing our accuracy bound work for a loss bound:

- Recall from Section 5.1 that we bound the accuracy by lower-bounding for each input pair $x, y \in G$ the *margin* $M_{x,y}$ with which the correct logit value exceeds any incorrect logit value in the original model. (This margin $M_{x,y}$ is computed as the difference between the idealized model's margin and the maximum logit difference between the original and idealized models.) For each input pair $x, y \in G$, we can guarantee that the original gets the correct answer on $x, y$ if $M_{x,y} > 0$; this results in a lower bound on accuracy. When considering loss, we can instead use translation-invariance of softmax to find that the contribution to the loss due to $x, y$ is

$$\mathcal{L}_{\text{ce}}(\boldsymbol{\theta}; x, y) \leq -\log \frac{e^{M_{x,y}}}{|G| - 1 + e^{M_{x,y}}}.$$

The average of these terms over all $x, y \in G$ is then an upper bound for the total cross-entropy loss.

- Another approach is to first use bi-equivariance to compute the true cross-entropy loss of the idealized model with a single forward pass and then to bound the $\ell_2$ norm between the idealized and original models' output logits using a variation of Lemma G.1. Combined with either the fact that cross-entropy loss is $\sqrt{2}$-Lipschitz (Lemma G.3) or with a inequality that takes into account second-degree information about cross-entropy (Lemma G.4), this gives a bound on the original model's loss.

In our experiments, we find that neither of these techniques suffices to give meaningful bounds on cross-entropy loss. This lack of success is somewhat to be expected—we start with approaches designed to bound accuracy, and then attempt to crudely adapt them to loss. Better bounds on loss would likely require new techniques which are out of scope for this paper. See Figure 9 for an example of loss bounds through the margin for models trained on $S_5$.

# K  BEYOND SYMMETRIC GROUPS

## K.1  OTHER GROUPS WITH REAL IRREPS

In the main text, we focus on the symmetric group $S_5$. For our purposes, this group is especially nice for several reasons:

- Symmetric groups $S_n$ have only real irreps, in the sense that every irrep over $\mathbb{C}$ is isomorphic to an irrep with only real matrix entries. See Section K.2 for a preliminary discussion of groups that do not have all real irreps.

- The minimum faithful $\rho$-sets of $S_n$ are small relative to the order of the group. In other words, $S_n$ has small faithful permutation representations because it can be embedded into itself $S_n \hookrightarrow S_n$. (In general, an arbitrary finite group $G$ can be embedded into a symmetric group $G \hookrightarrow S_n$ by Cayley's theorem, but unless $G$ itself is a symmetric group we must have $|S_n| > |G_n|$.)

- For groups of significantly smaller order, the training dataset is too small and we do not observe the grokking phenomenon. (Recall that training set size is proportional to $|G|^2$.) For groups of significantly larger

Related to the second point above, we empirically observe that groups with larger $\rho$-sets relative to group order are more prone to failure mode ($\rho$-bad), i.e. they typically miss a substantial portion of pairs in the double summation of Eq. 4. In this situation, we do not have a complete understanding of how the model attains high accuracy, and thus our bounds are correspondingly poor. Note that, although we cannot fully explain how neurons interact in this case, our per-neuron observations (B.2 1-3) hold for all finite groups we examine.

Despite these points, we are able to obtain nonvacuous bounds for models trained on the symmetric group $S_4$ (see Figure 10) and for models trained on the alternating group $A_5$ (see Figure 11).

## K.2  COMPLEX AND QUATERNIONIC IRREPS

An irrep being real is equivalent to it having positive Frobenius-Schur indicator $\iota(\rho) := |G|^{-1} \sum_{g \in G} \operatorname{tr}(\rho(g))$. In general, irreps have Frobenius-Schur indicator in $\{1, 0, -1\}$, corresponding to the irrep being *real*, *complex*, and *quaternionic* respectively. These three cases correspond to the ring of $G$-linear endomorphisms of the irrep being isomorphic to either $\mathbb{R}, \mathbb{C}, \mathbb{H}$. By Schur's lemma, the endomorphism ring is a real associative division algebra, so these are the only three cases.

In preliminary investigations of more general irreps $\rho$, we convert irreps over $\mathbb{C}$ with nonpositive Frobenius-Schur indicator to irreps over $\mathbb{R}$ of twice the dimensionality:

$$\tilde{\rho}(g) = \begin{bmatrix} \operatorname{Re} \rho(g) & -\operatorname{Im} \rho(g) \\ \operatorname{Im} \rho(g) & \operatorname{Re} \rho(g). \end{bmatrix} \tag{15}$$

We then find that the $\boldsymbol{A}, \boldsymbol{B}, \boldsymbol{C}$ matrices of Observation B.2(1) are approximately rank one when $\rho$ is real, rank two when $\rho$ is complex, and rank four when $\rho$ is quaternionic. In the complex case, we find also that when $\mathbb{R}^d$ is given the complex structure induced by Eq 15, the two singular vectors are conjugate, and correspond to equal singular values. Thus, we speculate that the neural network uses the same $\rho$-sets algorithm as in the real case, but over $\mathbb{C}$, and then takes the real part: letting $\rho \in \mathrm{GL}(n, \mathbb{C})$ and $\mathcal{B} \subseteq \mathbb{C}^d$ a finite $\rho$-set,

$$f_{\rho,\mathcal{B}}(z \mid x, y) = -\sum_{\boldsymbol{b},\boldsymbol{b}' \in \mathcal{B}} \mathrm{Re}\, \boldsymbol{b}^\top \rho(z) \boldsymbol{b}' \, \mathrm{ReLU}[\mathrm{Re}(\boldsymbol{b}^\top \rho(x)\boldsymbol{a} - \boldsymbol{a}^\top \rho(y)\boldsymbol{b}')].$$

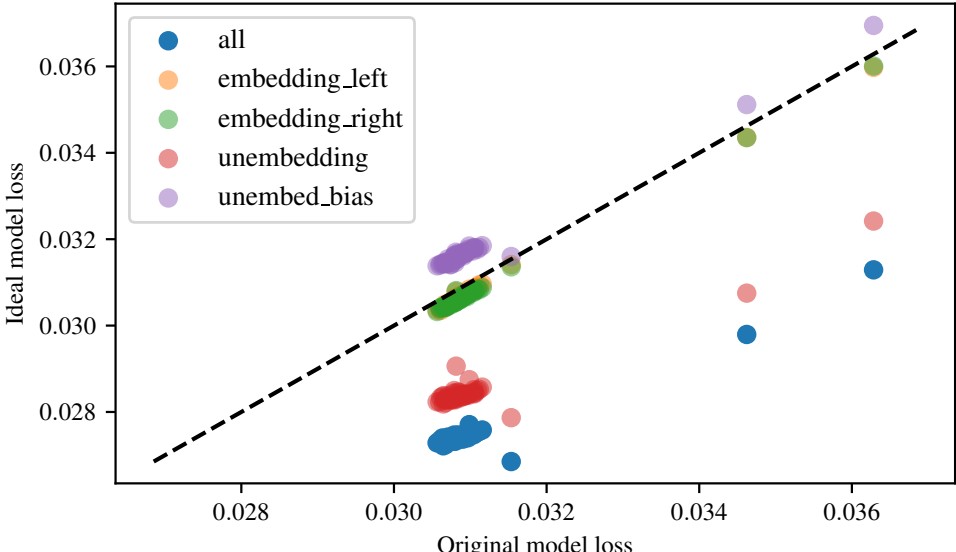

Figure 7: Cross-entropy loss of original model vs. cross-entropy loss of model with parameters partially exchanged for idealized version. 100 models trained on $S_5$, restricted to those where neither ($a$-bad) nor ($\rho$-bad). Legend indicates which parameters are exchanged; for instance, **red** points have unembedding weights $w_u$ swapped for idealized version. **Blue** points are the full idealized model. Note they have loss uniformly lower than original model. Points corresponding to left embedding are obstructed by those corresponding to right embedding.

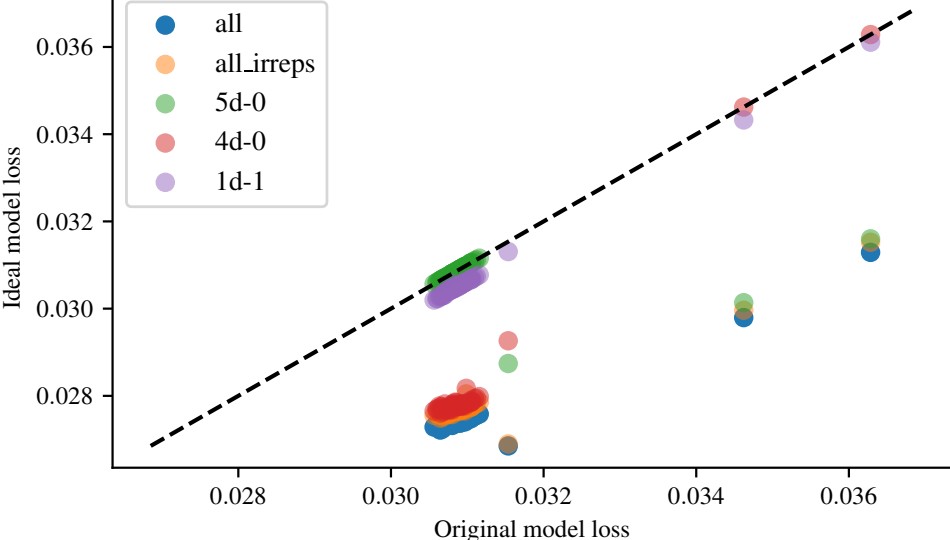

Figure 8: Cross-entropy loss of original model vs. cross-entropy loss of model with neurons (embedding and unembedding) corresponding to specified irreps exchanged for idealized version. 100 models trained on $S_5$, restricted to those where neither ($a$-bad) nor ($\rho$-bad). Legend indicates which parameters are exchanged. **Blue** points are the full idealized model while for **orange** points only neurons corresponding to *some* irrep are swapped (that is, dead neurons are preserved from the original). **Red** points correspond to the standard irrep 4d-0 and **purple** points correspond to the sign irrep 1d-1. See Section H for a full enumeration of irreps of $S_5$.

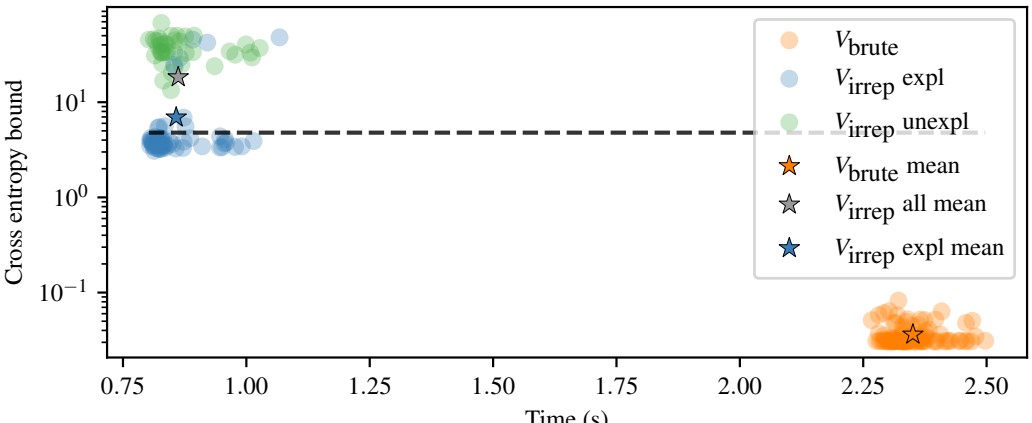

Figure 9: Cross-entropy bound vs. computation time for $V_{\text{irrep}}$ and $V_{\text{brute}}$ on 100 models trained on $S_5$. Points in **green** ($V_{\text{irrep}}$ unexpl) are models for which we find by inspection that our $\rho$-sets explanation does not hold, i.e. either ($a$-bad) or ($\rho$-bad); they make up 45% of the total. Points in **blue** are $V_{\text{irrep}}$ for explained models and points in **orange** are $V_{\text{brute}}$. **Black** dashed line is $\log|G| \approx 4.79$, the loss attained by a model that outputs uniform logit values. A priori, there is no guarantee that a given model does at least as well as the uniform baseline. Thus, in a sense, any finite upper bound on cross-entropy is nonvacuous.

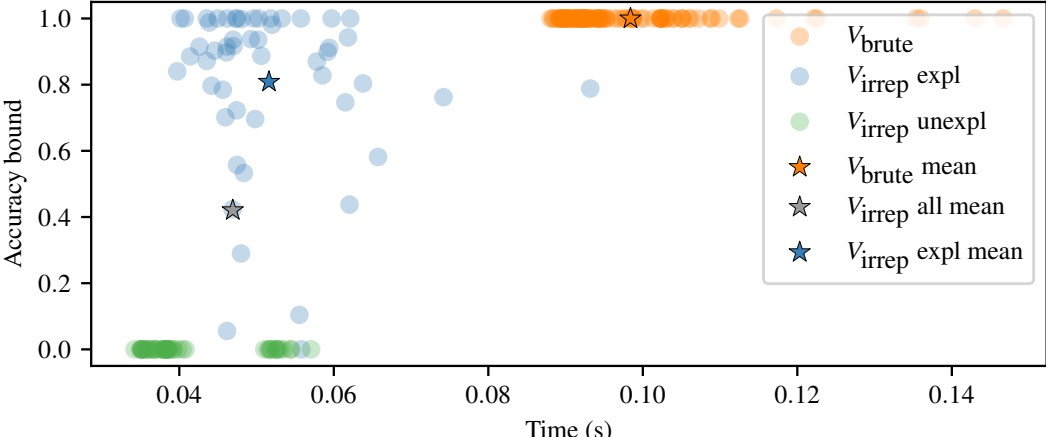

Figure 10: Accuracy bound vs. computation time for $V_{\text{irrep}}$ and $V_{\text{brute}}$ on 100 models trained on $S_4$. Points in **green** ($V_{\text{irrep}}$ unexpl) are models for which we find by inspection that our $\rho$-sets explanation does not hold, i.e. either ($a$-bad) or ($\rho$-bad); they make up 48% of the total. Note that the latter condition occurs much more frequently than for $S_5$. Points in **blue** are $V_{\text{irrep}}$ for explained models and points in **orange** are $V_{\text{brute}}$.

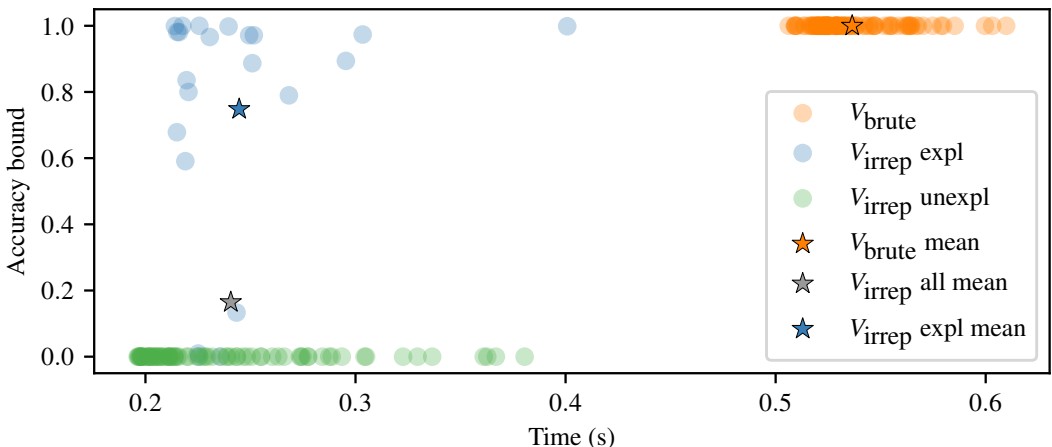

Figure 11: Accuracy bound vs. computation time for $V_{\mathrm{irrep}}$ and $V_{\mathrm{brute}}$ on 100 models trained on $A_5$. Points in **green** ($V_{\mathrm{irrep}}$ unexpl) are models for which we find by inspection that our $\rho$-sets explanation does not hold, i.e. either ($a$-bad) or ($\rho$-bad); they make up 78% of the total. Note that the latter condition occurs much more frequently than for $S_5$. Points in **blue** are $V_{\mathrm{irrep}}$ for explained models and points in **orange** are $V_{\mathrm{brute}}$.

