# OpenReview forum: "Towards a Unified and Verified Understanding of Group-Operation Networks"
_ICLR.cc/2025/Conference — ICLR 2025 Spotlight_

### Official Review · Reviewer_t8QQ · 2024-11-01

**Soundness:** 4
**Presentation:** 3
**Contribution:** 3
**Rating:** 8
**Confidence:** 3

**Summary:**

This paper unifies two previously proposed explanations for a small neural network trained in a controlled setting. Specifically, it studies the internals of a one-layer neural network trained to perform group composition on finite groups $G$. The input to the model is an ordered pair $(x, y)$ with $x, y \in G$, which are embedded as vectors, and the output is the group element. Prior work studied the same setting and proposed different explanations for the model behaviour: Stander et al. (2024) suggested that individual neurons develop a specialised coset behaviour where their left embeddings remain constant on right cosets and their right embeddings remain constant on left cosets, creating specific subsets $X_i$ where neuron activations sum to zero. Chughtai et al. (2023) found that each neuron operates in the linear span of matrix elements of some irreducible representation, implementing the matrix multiplication through ReLU nonlinearities and then maximising a trace computation to predict the group composition. First, for each of these explanations, the authors point out parts of the behaviour that is left unexplained. They then unify these explanations by showing that neurons are not just using irreducible representations randomly, but are organised in specific circuits. They also show exactly how the ReLU computation works through equation 4, which suggests that the function is bi-equivariant and is maximised when $z = x \star y“. Together these findings provide a more complete picture of the models computations.

To evaluate the quality of their explanation, they convert their understanding into a verifier program that aims to provide a compact proofs of model performance. Specifically, this verifier aims to use mechanistic insights to reduce the description length of the program (i.e. its runtime). They compare brute-force, the coset explanation (Stander et al.), and their own explanation in terms of accuracy bound and runtime over 100 models trained on the same task. They find that brute-force provides exact accuracy bounds but takes the longest to run (2.2 seconds), the coset explanation does not provide any non-vacuous accuracy bounds, and their explanation obtains a bound of 80 - 100 % for half the models and a non-vacuous bound of 0 % for the other half. Upon inspection, they find that the models where their explanation was not able to obtain non-vacuous bounds, the model converged to other solutions.

**Strengths:**

- The authors successfully reverse-engineer a neural network trained to perform group operations and provide a more complete explanation than prior work.
- They rigorously evaluate the quality of their explanation, highlighting that it only explains a subset of solutions a model with this architecture might learn in practice.
- Their evaluation exposes limitations of causal interventions as positive evidence of explanations.

**Weaknesses:**

Overall, I think this is a solid contribution without significant weaknesses.

**Questions:**

- You currently cite the survey of Geiger et al. (2024) for causal scrubbing (line 109). However, I believe it was first introduced in Chan et al. (2022; https://www.lesswrong.com/posts/JvZhhzycHu2Yd57RN/causal-scrubbing-a-method-for-rigorously-testing).

---

> ### Author Response · Authors · 2024-11-20
> **Response to reviewer t8QQ**
>
> Thank you for the overall supportive review of our work!
>
> > You currently cite the survey of Geiger et al. (2024) for causal scrubbing (line 109). However, I believe it was first introduced in Chan et al.
>
> We fixed this in the revised version.

---

> > ### Comment · Reviewer_t8QQ · 2024-11-26
> >
> > Thank you for your response. After reviewing the other feedback and the author responses, I will keep my rating.

---

### Official Review · Reviewer_ZpTx · 2024-11-03

**Soundness:** 4
**Presentation:** 2
**Contribution:** 3
**Rating:** 8
**Confidence:** 4

**Summary:**

- The paper presents an algorithm by which a one hidden layer MLP network (with embeddings) does composition in $S_5$
    - The algorithm is somewhat involved, but the key step is constructing an expression for $f(z | x,y)$ (eqn 4) that is purely a function of $x^{-1}zy^{-1}$ and maximised at $e$.
    - This is constructed in an interpretable way using $\rho$-sets, an introduced concept of a set of vectors permuted by irreps of the group G. For a set of $k$ vectors, it needs $k^2$ neurons (one corresponding to each pair of $\rho$-set vectors).
    - The full network consists of several such constructions, in parallel, using disjoint sets of neurons and possibly different irreps and $\rho$-sets
- This is the same setting as studied by previous mechanistic interpretability works Chughtai et al & Stander et al, but presents a much more detailed algorithm, with a clear story for each parameter and the role of each neuron.
    - The story provided here helps unify and clarify observations in each paper such as coset concentration and irrep sparsity.
    - Some aspects of prior work are criticised, such as the specific causal interventions used as evidence in Stander et al
- This explanation is used to provide a lower bound on the model's accuracy, by analysing the margin (correct logit - max incorrect logit) in the idealised model, and comparing that to the worst case logit deviation in the real model.
    - This accuracy bound is proven via a scheme that takes asymptotically less time than brute force trying every input (about 3x faster in practice). This proof fails on some networks, but this seems to correspond to ones where the explanation is imperfect.
    - This is referred to as a compact (i.e. short) proof, inspired by Gross et al
    - The existence of a compact proof seems to be taken as evidence that the explanation is correct, as it gives a faster yet provably valid verification method

**Strengths:**

1. The authors have found an elegant yet highly non-obvious algorithm for group composition in a single ReLU layer, that multiple prior papers missed. This is a valuable contribution to the literature.
    - Further, the provided explanation clarifies and adds useful context to observations made in prior work
2. The presented compact proofs are highly detailed and rigorous, and actually explicitly go through every key detail, rather than hand-waving annoying points.
3. It demonstrates compact proofs as an interpretability metric in a much more complex setting than prior work (Gross et al)
4. The authors do a good job of highlighting weaknesses, e.g. the times the proof does not work, the failed V_coset proof, and clearly reporting the actual runtimes of the proofs
5. Though highly technical and complex, to the best of my ability to tell, the maths largely checks out. I did not follow the fine detail of all of the appendices.

**Weaknesses:**

1. This is only studied on $S_5$
2. The compact proof is only 3x faster than brute force
3. The paper is highly technical and at times quite difficult to follow, especially as it builds deeply on 3 prior papers! Though the authors have clearly made an effort to be clear, and this is an inherently complex work. This took me significantly longer than other reviews.
4. The link between finding a compact proof of a bound on accuracy, and verifying a mechanistic explanation, seems somewhat unclear

**Questions:**

# Major Comments
*The following are things that, if adequately addressed, would increase my score*
1. The key thing that would improve this paper, in my opinion, is making it clearer, especially key technical details.
    - I found sections 4 and 5 difficult to follow - the concepts of coset concentration and irrep sparsity are introduced, without much motivation, and are then not necessary to explain the algorithm. I would personally reverse the flow of 4 & 5.1 and work backwards:
        1. Begin with Eqn 4, and observe that this is bi-equivariant, and sometimes maximised at e (citing lemma F.3)
        2. Show that this is equivalent to Eqn 3
        3. Observe that each term in the sum can be a neuron, and what the relevant w_r, w_l, and w_u need to be for that - we've now constructed a valid algorithm!
        4. Irrep sparsity and coset concentration can then be explained as prior observations, and shown to follow from this algorithm.
    - I find the term "compact proof" quite cryptic, and it took me until about page 7 to figure out what was going on. A decent part of the confusion is that you use proof to refer to what I'd normally call a program. I would have benefitted a lot from an intuitive explanation in the intro or start of section 6 (ideally both, plus something in the abstract). Something like:
        - A guarantee of model accuracy is a program that can be run to guarantee that models will always give the correct answer at least X% of the time, for some X (i.e. lower bound its accuracy). This can always be done for the brute force "try every possible input" program, but it seems that mechanistic understanding of a model should enable more efficient programs. These are referred to as compact proofs. The efficiency of the program, and closeness of the accuracy bound to the true accuracy, can be taken as metrics of the quality of our explanation.
2. I'm somewhat skeptical of compact proofs as an interpretability metric, for several reasons - I would love to be convinced otherwise though:
    - They bound accuracy, not loss. But a model's performance can be over-determined, with several parallel components each being sufficient to ensure perfect accuracy but all needed to recover the loss, as is common in toy systems like this (you do need to be able to bound the effect of other components, but not necessarily to understand them). IMO an explanation that doesn't understand all such components is incomplete, but it may get fantastic accuracy.
    - It's not clear to me that a mechanistic explanation, even if extremely accurate, should always enable faster proofs. Or even be robust to worst case guarantees at all.
        - While it did in this work, this was *extremely* specific to the setting and explanation, and I don't feel confident there would be other approaches for less mathematically elegant algorithms.
    - The framing in the paper was that being asymptotically faster than brute force was the key thing. But in practice, the coefficient on the compact proof was much worse, and it was 3x faster not 120x faster. IMO 3x is the relevant number here.
    - That said, I find the fact that they seem to identify networks where your explanation is incomplete to be quite compelling.
3. Similarly, I would be excited to see other evidence that your explanation is correct - it makes a lot of predictions about the form of the parameters and activations!
    - Can you learn a and a $\rho$-set that explains a cluster of neurons? Does it have size $k^2$ exactly?
    - How well does your prediction for a neuron's activations match it in practice? What's the MSE and correlation? If you replace the neuron with the prediction (either one at a time, or on the full group for one $\rho$-set) what happens to model performance.
    - How close are the model's parameters (or at least, $w^i_l$ etc) to the predicted form? What if you substitute part of those?
    - I found the claim in Lines 502-503 that causal interventions can only yield weaker positive evidence to be overstated - this applies to the interventions used in Stander et al, but IMO those are pretty weak interventions and not that compelling by the standards of current mechanistic interpretability. This doesn't mean there don't exist more compelling causal interventions.
4. Separately, I'm extremely skeptical that compact proofs will scale beyond very toy networks (let alone to frontier systems). Being robust to worst case scenarios on the entire space of inputs seems highly unrealistic to me for eg a language model or image model. Methods don't have to scale to be interesting, of course, but this limits my excitement about the method. I'd value arguments for why the method might extend to eg imperfect coverage of the input space.
5. I'd be very curious to see how well "the square of the size of the smallest $\rho$-set$" correlates with the probability that $\rho$ is learned, eg using the numbers in Chughtai et al. This would significantly clarify the results in Chughtai et al re universality if true.

# Minor Comments
*The following are unlikely to change my score, but are comments and suggestions that I hope will improve the paper, and I leave it up to the authors whether to implement them. No need to reply to all of them in the rebuttal*
1. Line 243: Explicitly note that we can set w^i_l(x) to whatever we want, it's a lookup table. It's a construction, not a conclusion. This confused me at first
2. I find the term "compact proof" somewhat confusing. To me, proof connotes showing something rigorously about abstract mathematical objects . But perhaps this is just taste, and this notion of proof is common in e.g. the field of formal verification?
3. I don't understand what Figure 1 is trying to show, a shame as you clearly put in effort there! How is S3 mapped to points on a hexagon? What are the terms in the top row with 4 vertices circled? What does adding them mean? What is X_12? Etc I recommend significantly clarifying or changing the figure
4. Obviously, it would be great to replicate the paper's results on other groups! A5, A6, S4 seem natural to try. I predict the results to hold up though.
5. I find the definition of compact proofs somewhat odd - what exactly does it mean to be a valid lower bound for *any* explanation string? It strikes me as odd that your compact proof must first begin by eg verifying if a subset of G is a subgroup, when that seems independent of the model.
6. I personally find Eqns 2 and 3 clearer by replacing $b^T \rho(x)a$ with $a^T \rho(x^{-1}) b$ (which is equal since the transpose of a scalar is the identity), as this motivates the subtraction and substitution more clearly.
7. I liked the point in the appendix that $ReLU(x) = (x + |x|)/2$, though it was not clear to me why the $x$ part cancelled out.
8. Line 109: Causal Scrubbing was introduced in [Chan et al](https://www.alignmentforum.org/posts/JvZhhzycHu2Yd57RN/causal-scrubbing-a-method-for-rigorously-testing), not Geiger et al
9. Line 116: It would be good to define the word equivariance here
10. Line 194: I found the claims about embeddings being constant on a coset for a neuron very confusing. It felt like it was asserting this about *all* networks, not explaining a property of a specific constructed network
11. Line 196: What does it mean for a subset of G to be common to a family of cosets? Cosets are subsets of G, so surely the intersection of a family of cosets is a set of subsets, not a subset?
12. Line 436: Why do you refer to neurons as functions $G\to \mathbb{R}$ rather than $G^2\to\mathbb{R}$? They have two inputs, $x$ and $y$, right?
13. Lemma F.3: Are you arguing that all $\rho$ have the decomposition described here? If not, which ones do, and does this correspond to the irreps learned by the model?
14. In table 2 in Appendix G, why does the minimal $\rho$-set size go above 5? Naively, it feels like an irrep of S5 should always be able to permute a set of 5 vectors.
15. Line 452: The claim that the frequency $\rho$ is learned correlates with its minimal $\rho$-set size seems questionable to me. The less frequent 4D one and more frequent 5D one are learned about the same amount of the time in Chughtai et al (Figure 7), despite having $\rho$-set size 10 and 6 respectively, and both less than half as often as the more frequent 4D one (minimal size 5). Your finding may help explain which of the representations at a given dimension is chosen, but it's clearly incomplete

---

> ### Author Response · Authors · 2024-11-20
>
> First of all, many thanks for an extremely thorough and insightful review. We greatly appreciate the effort.
>
>
>
> ### Major comments
>
> > 1.1 The key thing that would improve this paper, in my opinion, is making it clearer, especially key technical details. I found sections 4 and 5 difficult to follow - the concepts of coset concentration and irrep sparsity are introduced, without much motivation, and are then not necessary to explain the algorithm. I would personally reverse the flow of 4 & 5.1 and work backwards:
>
> We have rewritten section 5.1 and started with (Eq 4) to make the final result clearer. We think it is better to keep 4 & 5.1 seperated, to clearly distinguish between our work and previous work. Section 4 is also not necessary to understand our circuit, but helps clarify how our approach unifies the previous ones.
>
> > 1.2 I find the term "compact proof" quite cryptic, and it took me until about page 7 to figure out what was going on. A decent part of the confusion is that you use proof to refer to what I'd normally call a program. I would have benefitted a lot from an intuitive explanation in the intro or start of section 6 (ideally both, plus something in the abstract). Something like:
>
> Thanks for this suggestion. We agree that the introduction was vague about this concept and we have added a slightly modified version of your suggestion in the introduction.
>
> > 2. I find the term "compact proof" somewhat confusing. To me, proof connotes showing something rigorously about abstract mathematical objects . But perhaps this is just taste, and this notion of proof is common in e.g. the field of formal verification?
>
> What we refer to as compact proofs can be thought of as proofs of statements about abstract mathematical objects (the model weights) in the standard mathematical sense. A compact proof of a model $M$ consists of two parts:
> - Let $W$ be the space of weights and $D$ the space of inputs (let's say uniformly sampled, but you could also try to prove statements for a different distribuition). Then for a map $C:W\to\mathbb{R}$ (depending on the interpretation), we prove an inequality $$ C(w)\leq \mathbb{E}_{x\in D}[Acc(w,x)]$$
> i.e. it is a sound lower bound for the accuracy. In our case the map $C$ is "run the algorithm in Appendix E and return the final number you get in Step 7". Note that the proof of this inequality does not depend on model or dataset size.
>
> - Use the model weights to calculate $C(w)$. This can be thought of as proving that $C(w)$ is what it is, with length of proof equal to the execution time. The bulk of the proof's length is this step.
>
> But indeed we also refer to a definition of compact proofs that is common in formal verification (though we don't find it necessary to think in these terms to understand it): A fully formal proof in logic is a tree where each node is the application of an axiom of the theory.  (In the dependent type theories used by, Coq, Agda, Lean, etc., a fully formal proof is a well-typed abstract syntax tree of the theory.). By "compact" we just mean "short", i.e., the number of nodes in the tree is small (or, in dependent type theories, where the worst-case time to check validity grows as roughly the busy-beaver number of the size of the AST, we mean that the proof-checking time is short).
>
> > I'm somewhat skeptical of compact proofs as an interpretability metric, for several reasons - I would love to be convinced otherwise though:
> >2.1 They bound accuracy, not loss.
>
> Our current strategy to generate compact proofs works by bounding the margin of the logits. This strategy can be modified to instead bound the cross-entropy loss. However, this adaptation to a loss bound is rather crude and results in fairly weak bounds. This isn't an inherent limitation of the compact proofs framework; rather, our paper was simply more focused on accuracy than on loss. See Appendix J for details.

---

> > ### Author Response · Authors · 2024-11-20
> >
> > ### Major comments contd
> >
> > > But a model's performance can be over-determined, with several parallel components each being sufficient to ensure perfect accuracy but all needed to recover the loss, as is common in toy systems like this (you do need to be able to bound the effect of other components, but not necessarily to understand them). IMO an explanation that doesn't understand all such components is incomplete, but it may get fantastic accuracy.
> >
> > There are two cases. Either it is the case that:
> >
> > 1. The component you do understand is so much stronger than all other components that even if the other components were broken, the model would still behave correctly, and if the component you do understand were reversed, the other components would not be enough to make up the difference.  Note that in this case, we will be able to bound loss as well as accuracy even without "fully understanding" the model, and we would claim that, while we lack complete understanding of "the exact behavior of the model on the dataset", we do have complete understanding of "how the model does as well as it does on the dataset".
> >
> > 2. The component you do understand is only sufficient to guarantee accuracy in the case that the other parallel components do not harm the output (and in fact they improve the output).  Here the case is weaker, but it is still the case that a compact proof must provide an explanation of how it is the case that these other components do not get in the way. To get a sufficiently compact proof, we claim, you must understand the component pretty deeply, though you don't necessarily have to understand how they contribute positively as opposed to just how they don't harm.  But indeed a tight bound on loss constrains the explanation differently than a tight bound on accuracy, and you can vary the theorem statement to get interpretability metrics on different explanation targets.  We see this as a feature (the metric can be customized to account for variation in what we are trying to explain about the model) rather than a bug.
> >
> >
> >
> > > 2.2 It's not clear to me that a mechanistic explanation, even if extremely accurate, should always enable faster proofs. Or even be robust to worst case guarantees at all.
> > While it did in this work, this was extremely specific to the setting and explanation, and I don't feel confident there would be other approaches for less mathematically elegant algorithms.
> >
> > Indeed this is currently an empirical question, and this paper is a bit more evidence in favor.  But let's separate two points here: (1) Do (mechanistic) explanations correspond to proofs?  (2) Does quality of explanation correspond to tightness of bound and length of proof?
> >
> > (1) Seems to be something of an empirical / philosophical question, and the biggest point of divergence is indeed the worst-case vs average-case distinction regarding the model's weights. We discuss the worst-case vs average-case distinction regarding the data in reponse to major comment 4.
> >
> > The argument in support of (2) is just that better explanations are better compressions, either by being less lossy or by giving higher compression ratios.  Insofar as this property seems true of explanations in general, it should also apply to compact proofs, insofar as (1) is true.
> >
> >
> >
> > > 2.3 The framing in the paper was that being asymptotically faster than brute force was the key thing. But in practice, the coefficient on the compact proof was much worse, and it was 3x faster not 120x faster. IMO 3x is the relevant number here.
> >
> > The speed of running the proof serves as a proxy for the metrics we actually care about: FLOPs or the computational complexity of the model. We believe that as we increase the the size of the group these differences become more dominant. For example, fixed costs of running the program will become less significant and we expect the speed up factor to go up. Unfortunately, we lack the resources to train models on significantly larger groups. (Recall that, per epoch, training time is proportional to $|G|^3$.)
> >
> >
> >
> > > 2.4 That said, I find the fact that they seem to identify networks where your explanation is incomplete to be quite compelling.
> >
> > We agree and found this to be a compelling argument in favour of this approach!
> >
> > >3. Similarly, I would be excited to see other evidence that your explanation is correct - it makes a lot of predictions about the form of the parameters and activations!
> >
> > We added several experiments that you suggested. See Appendix C "Additional evidence for $\rho$-set circuits" and the figures referenced in there.

---

> ### Author Response · Authors · 2024-11-20
>
> ### Major comments contd 2
> >Can you learn a and a $\rho$-set that explains a cluster of neurons?
>
> This is precisely how we arrive at bounds on accuracy. We explicitly write down $\rho$-set circuits (the idealized model) and then bound their distance in output space from the original model.
>
> >Does [a cluster of neurons corresponding to a $\rho$-set] have size $k^2$ exactly?
>
> This is typically true, with two caveats:
> - Sometimes (as briefly mentioned in footnote 9 of the revised text), there are more than one neuron corresponding to a single pair of the double summation. These "duplicate neurons" are a minor technical detail that can be dealt with easily as long as the sum of magnitudes of all neurons corresponding to each pair is uniform across all pairs (Observation B.2.7). Empirically this is indeed the case
> - More importantly, for some models there are no neurons corresponding to a substantial number of pairs in the double summation, i.e. the number of neurons in the circuit is much less than $k^2$; this failure mode is labeled ($\rho$-bad) in the revised text. In this case we do not fully understand the model's performance, and correspondingly we are unable to obtain good bounds. (If there are only a handful of missing neurons, we can simply add them to the idealized model and bound the discrepancy in output logits due to these neurons.)
>
> >3.3 I found the claim in Lines 502-503 that causal interventions can only yield weaker positive evidence to be overstated - this applies to the interventions used in Stander et al, but IMO those are pretty weak interventions and not that compelling by the standards of current mechanistic interpretability. This doesn't mean there don't exist more compelling causal interventions.
>
> We agree and modified this sentence to make a more careful claim.
> Indeed the main thing lacking in casual scrubbing is not the strength of evidence in either direction (there's a sense in which casual scrubbing can be seen as a sampling-based proof), but rather an adequately developed notion of compactness that can be used to evaluate the quality/depth of the explanation (the brute force explanation is the best, as far as causal scrubbing is concerned).
>
> > 4. Separately, I'm extremely skeptical that compact proofs will scale beyond very toy networks (let alone to frontier systems). Being robust to worst case scenarios on the entire space of inputs seems highly unrealistic to me for eg a language model or image model. Methods don't have to scale to be interesting, of course, but this limits my excitement about the method. I'd value arguments for why the method might extend to eg imperfect coverage of the input space.
>
> It's worth noting that proofs are worst-case over the unexplained portion of the *model weights* but average case over the *input distribution*.  We can easily deal with imperfect coverage of the input space by simply discarding that portion of the input space from the accuracy bound, resulting in either 1) a decreased accuracy bound for the entire space or 2) a less compact proof, if the discarded portion is dealt with another way, say using brute force. Chernoff's concentration inequality can be used to get less crude bounds on the "typical" case behavior w.r.t. inputs (there is work in progress on this).
> Additionally, ARC Theory's work on [surprise accounting](https://www.alignment.org/blog/formal-verification-heuristic-explanations-and-surprise-accounting/) in heuristic arguments can be seen as a way to generalize compact proofs from worst-case over model weights to typical-case over model weights.
>
> We made some more comments about the compact proofs approach in our general response.
>
> >5. I'd be very curious to see how well "the square of the size of the smallest rho-set correlates with the probability that rho is learned,” eg using the numbers in Chughtai et al. This would significantly clarify the results in Chughtai et al re universality if true.
>
> The statement we intended to make is: The order of frequencies in Figure 7 of Chughtai et al. is the same as the ordering of the rho-set size in Table 3. We have clarified this in section 7.1 and hope this also addresses minor comment 15.

---

> ### Author Response · Authors · 2024-11-20
>
> ### Minor comments
>
> >Minor comments
>
>
>
> > 3. I don't understand what Figure 1 is trying to show, a shame as you clearly put in effort there! How is S3 mapped to points on a hexagon? What are the terms in the top row with 4 vertices circled? What does adding them mean? What is X_12? Etc I recommend significantly clarifying or changing the figure
>
> Thanks for the input. The points on the hexagon are one way to arrange elements of $S_3$. The way it is done is not very important, the relevant part is that the group $S_3$ acts on this via reflections/rotations (and other kinds of symmetries). In our example, right multiplication by $(123)$ acts via rotation by $120^\circ$.
> We decided to remove the upper part. We appreciate any feedback, if you think it would make the figure easier to understand.
>
> >4. Obviously, it would be great to replicate the paper's results on other groups! A5, A6, S4 seem natural to try. I predict the results to hold up though.
>
> We have extended Appendix K with results for A5 as well as some discussion of difficulties we face in applying our interpretation to other groups. Unfortunately, we are fairly restricted in the size of groups we can train models for --- for large groups like A6, we lack the computational resources (training cost per epoch is proportional to $|G|^3$), while for small groups like S4, there are too few training points and we do not observe the grokking phenomenon at all.
>
> >5. I find the definition of compact proofs somewhat odd - what exactly does it mean to be a valid lower bound for any explanation string? It strikes me as odd that your compact proof must first begin by eg verifying if a subset of G is a subgroup, when that seems independent of the model.
>
> The interpretation string / verifier setup is introduced in order to correctly formalize the notion of an interpretation corresponding to a *specific model* rather than all models. Thus, the interpretation string is allowed to vary with the model being interpreted. On the other hand, we cannot allow the verifier to vary with the model -- if it could, for each model, we could trivially define the verifier to output the model's true accuracy in constant time.
>
> For similar reasons, we do not allow the verifier to vary over the group being trained on. If it did, we could again construct an (asymptotically) trivial solution by, say, memorizing a lookup table of all models that attain good accuracy on a specific group up to some precision.
>
>
>
> >11. Line 196: What does it mean for a subset of G to be common to a family of cosets? Cosets are subsets of G, so surely the intersection of a family of cosets is a set of subsets, not a subset?
>
> We mean that the subset $X\subseteq G$ is a member of both families, i.e. is an element of their intersection (which indeed is a set of subsets).
> We edited this part and hope it is clearer now.
>
> >12. Line 436: Why do you refer to neurons as functions G->R rather than G->R^2? They have two inputs, x and y, right?
>
> We decompose the pre-embedding part of the MLP into a left and right part with corresponding left and right neuron, as defined in Section 3.2.
>
> >Lemma F.3: Are you arguing that all have the decomposition described here? If not, which ones do, and does this correspond to the irreps learned by the model?
>
> Generally, this is not true. This lemma is true in case that H\G/H has two elements, which applies to our situation of H=S4 and G=S5. (It might be the case that the lemma is true in more general cases). We added a comment below Lemma F.3.
>
> Note that the validity of the compact proof does not depend on this lemma holding. The proof instead leverages bi-equivariance to verify that the output is maximized at the correct logit by computing a single forward pass. In principle, in cases where the lemma holds, we can prove model accuracy using *zero* forward passes, but we do not do this.
>
> >14. In table 2 in Appendix G, why does the minimal -set size go above 5? Naively, it feels like an irrep of S5 should always be able to permute a set of 5 vectors.
>
> Given a permutation representation $f:G\to S_n$, we can consider $f$ as a degree-$n$ linear representation of $G$ consisting of permutation matrices. There exists a $\rho$-set corresponding to this permutation representation (and thus a $\rho$-set of size $n$) if and only if $\rho$ is present in the decomposition of $f$ into irreps.
>
> For instance, if $G=S_5$ (and $f=id$), the corresponding linear representation admits two subrepresentations: the one-dimensional trivial irrep and the four-dimensional standard irrep. It's therefore impossible for e.g. the other four-dimensional irrep to have a minimal $\rho$-set size of $5$.

---

> ### Comment · Reviewer_ZpTx · 2024-11-25
> **Response to rebuttal (1)**
>
> Thanks to the authors for their detailed responses and improvements. My biggest concern was lack of clarity. I appreciate the improvements, but this is still a fairly complex paper and I think there's significant room for improvement - I detail my thoughts in another comment for space.
> - Compact proofs:
>     - Thanks for your explanation of how compact proofs can be used on loss, I find this persuasive (at least that it's not a theoretical flaw - I won't believe that it's solvable in practice until I see good empirical demonstrations of this)
>     - "better explanations are better compressions" While I agree with this, I do not agree that this implies that good explanations lead to better compact proof generation algorithms. Compact proof generation algorithms seem like a fairly peculiar thing, and like they eg need to exploit specific symmetries in the model to work, which I don't think necessarily need to be part of a good explanation
>     - "We believe that as we increase the the size of the group these differences become more dominant." Sure, but this paper only provides empirical evidence on S5, and you provide no evidence that the algorithm works on eg S4 or S6. I acknowledge that S6 or S7 are a pain to train, but I think that given that you only provide evidence that your explanation is correct on a single group size, the empirical speedup is a better quantity than the asymptotic value.
>     - Thank you for the clarification that it's worst case over model weights not input space, this is a fair point
> - Empirical evidence:
>     - Thanks for these, I find Figures 5, 7 and 8 to be convincing evidence that your interpretation is correct.
>     - Nit: It'd be good to say that you find the rho, B and a via the algorithm in appendix B.3
> - Minor comments:
>     - I disagree with your statement in line 511 that causal interventions do not provide a precise notion of explanation quality - various metrics like faithfulness and completeness are used in the circuit finding literature. I'm sympathetic to complaints that these are bad metrics, but imprecise seems the wrong criticism. Most of my criticisms would be that we don't know exactly what we're measuring or how to set it up right, but it seems that similar is true for compact proofs
>     - "We see this as a feature (the metric can be customized to account for variation in what we are trying to explain about the model) rather than a bug." I think this is a reasonable statement, but worth explicit discussion in the paper (or appendix if you remain highly space constrained). It sounds like compact proofs *if* used correctly, can be a flexible tool, but like it has a bunch of footguns if you don't set up the problem correctly, and get less interesting results than expected. This seems worth warning/instructing readers about
> - My overall take is that this is an interesting paper that covers a lot of ground and does meaningful theoretical work. I'm personally not particularly optimistic about proof-based approaches to interpretability, but I'm happy to see careful and rigorous work making progress here, like this paper. I do still have concerns about this paper, notably around it being hard to read to people without significant background in this area. **I will increase my score to an 8, as I consider this paper to be a meaningful contribution to the literature that I would be sad to see rejected, but would give it a 7 if that was an option, as there's still significant room for improvement on clarity of writing**

---

> ### Comment · Reviewer_ZpTx · 2024-11-25
> **Response to rebuttal (2) - Clarity improvements**
>
> To evaluate the clarity, I've tried to go through the revised paper pretending I was seeing it for the first time, and still find it dense and unclear:
> - Equivariance: I wouldn't know what you meant by equivariance given the description in the abstract & intro, despite this seeming a crucial contribution (I am familiar with the word, but not what it meant here). I recommend adding a 1 sentence definition to the intro - the formal definition in line 239 would have clarified things a lot.
>     - It would also be good to state in the abstract that this focuses on S5 specifically
> - Compact proofs: I would not understand what it means in the abstract. In the introduction the prose is significantly improved, but I expect I would still be confused. In particular, when I see program on the model my mind jumps to "weird wording for running the model on some input" and when I see formal proof I imagine "something a human wrote", through referring to brute force immediately after helps clarify. More broadly I find the whole concept fairly unintuitive - you're using an explanation to provide guarantees on the model's performance, but the guarantees don't actually require or assume an explanation, it's that the explanation  motivates a guarantee creation process, and the metric is "fraction of inputs on which we can guarantee correctness" and think it would benefit from more exposition
> - A narrative that would feel clearer to me would be something like the below - is this correct?
>     - When we have a precise mechanistic explanation of a model, we would like to rigorously show that this works.
>     - The real model will differ somewhat from this ideal explanation, due to noise or imperfections in our analysis
>     - To rigorously validate our analysis, we would like to bound this deviation, in a way that gives us formal guarantees, not just approximations. We focus on guarantees of the form "on X% of all possible inputs, the model gets the right answer"
>     - A formal guarantee essentially looks like a (potentially extremely long) mathematical proof, typically produced by an algorithm not a human.
>     - The simplest formal guarantee is to brute force try every possible input to the model - this works on any model, does not require a mechanistic explanation, and is a perfectly tight bound.
>     - But we believe that a mechanistic explanation should let us produce a formal guarantee via a faster algorithm than just trying every possible input, eg by exploiting symmetries predicted by the explanation. We hope that we can get a speedup while still finding a fairly tight bound. The resulting algorithm can be run on any model, but will only produce good guarantees on models with the properties predicted by the explanation. Being able to provide an efficient guarantee on model's performance therefore provides strong evidence for the correctness of our explanation.
> - I still don't really get figure 1 - how do x and y correspond to what's on the hexagon? How did you map those elements of S3 to those points? Why is (123) a rotation by 120 degrees? Presumably (12) is not a rotation of 60 degrees, since it has order 2. Currently this figure adds negative clarity for me
> - Section 4:
>     - I understand the desire to keep prior work and your work separate, but I think that swapping section 5 and section 4 (or moving section 4 to an appendix) would significantly improve clarity. Section 4 might add value to a reader familiar with Chughtai et al and Stander et al, but are needless complication to those who aren't (and you can add a reference to the section on prior work at the start of the section on your work).
>     - As is, the paper reads like "a review of concepts/empirical observations in prior work, and why you believe them to be limited and having a bunch of degrees of freedom", which the reader must try to understand and keep in their heads, followed by your algorithm, which, as far as I can tell, doesn't require the reader to understand the prior work at all.
>     - If the goal is to emphasise how your paper makes a contribution going beyond past work, I think this is still done well by swapping the section, as you can then explain how each observation follows from your algorithm, and point out all the details left unspecified by prior work that you fill in.
>     - Nit: I think it would also help to emphasise that coset concentration and irrep sparsity are empirical, approximate observations, not mathematical properties of the network - wording in line 188 like "left embeddings are constant" feels too strong
> - Section 5: Beginning the section with equation 2 is a significant improvement! Thank you
>     - It would help to define a,b,b',B immediately after the equation - the reader should be able to understand what this equation means without needing to read the next several paragraphs
>     - Nit: It would be good to say that the logits are (approximately) a linear combination of such terms

---

> > ### Author Response · Authors · 2024-11-29
> >
> > Thank you for your additional comments! We've found them very helpful in improving the clarity of our paper.
> >
> > >  I do not agree that this implies that good explanations lead to better compact proof generation algorithms. Compact proof generation algorithms seem like a fairly peculiar thing, and like they eg need to exploit specific symmetries in the model to work, which I don't think necessarily need to be part of a good explanation
> >
> > One presumably needs to make use of some kind of structure found in the model weights in order to explain them compactly. In our case we found and and used a fairly strict symmetry, but for other settings maybe broader and less restrictive notions of symmetry/structure could be leveraged. Whether this can actually be done for more complex settings is an important empirical question for future work.
> >
> > > this paper only provides empirical evidence on S5, and you provide no evidence that the algorithm works on eg S4 or S6 [...] the empirical speedup is a better quantity than the asymptotic value.
> >
> > We state the empirical speedup in the abstract. We also added results for S4. (We increased the portion of the input space used in the training set to 80% in order to induce grokking for this group.)
> >
> > > Nit: It'd be good to say that you find the rho, B and a via the algorithm in appendix B.3
> >
> > Fixed
> >
> > > I disagree with your statement in line 511 that causal interventions do not provide a precise notion of explanation quality
> >
> > Fixed. We hope the current phrasing more accurately conveys our point.
> >
> > > It sounds like compact proofs if used correctly, can be a flexible tool, but like it has a bunch of footguns if you don't set up the problem correctly, and get less interesting results than expected. This seems worth warning/instructing readers about
> >
> > We added a sentence to Appendix J warning about this (line 1388).
> >
> > > Equivariance: I wouldn't know what you meant by equivariance given the description in the abstract & intro
> >
> > We add a sentence to the introduction briefly explaining equivariance (line 49).
> >
> > > Compact proofs: I would not understand what it means in the abstract. In the introduction the prose is significantly improved, but I expect I would still be confused
> >
> > > A narrative that would feel clearer to me would be something like the below - is this correct?
> >
> > We agree with your narrative and we've incorporated it into the revised introduction.
> >
> > > I still don't really get figure 1 - how do x and y correspond to what's on the hexagon? How did you map those elements of S3 to those points? Why is (123) a rotation by 120 degrees? Presumably (12) is not a rotation of 60 degrees, since it has order 2.
> >
> > We agree that the original figure may have been more confusing than illuminating; thus we removed it and replaced it with a plot of 3d irreps for S4 and A5. (S5 does not have any 2d or 3d irreps, unfortunately.) To answer your question: the original figure was meant as a cartoon depiction of a higher-dimensional space, specifically the 4(=2x2) dimensional space inhabited by the matrices of the standard 2d irrep of S3. Thus the geometry of the hexagon and the 120 degree rotation were somewhat arbitrary choices that did not correspond to anything precise.
> >
> > >  I think that swapping section 5 and section 4 (or moving section 4 to an appendix) would significantly improve clarity.
> >
> > We re-ordered the sections as suggested.
> >
> > > Nit: I think it would also help to emphasise that coset concentration and irrep sparsity are empirical, approximate observations, not mathematical properties of the network - wording in line 188 like "left embeddings are constant" feels too strong
> >
> > We edited the section to say "approximately constant" etc.
> >
> > > It would help to define a,b,b',B immediately after the equation
> >
> > > Nit: It would be good to say that the logits are (approximately) a linear combination of such terms
> >
> > Done

---

> ### Comment · Reviewer_ZpTx · 2024-12-01
>
> Thanks a lot for the updates! I think the clarity has been substantially improved, and am **happy to "update" my score from a lukewarm 8 to a wholehearted 8**. I'm excited about the paper, and think this is a useful contribution to the literature.
>
> Some notes:
> * For the contributions in the introduction, contribution 1 should probably be moved to 2 or 3, I think, since it's now deprioritised to section 6.
> * For the camera ready, it might be nice to add a more informal appendix giving advice to researchers trying to use compact proof-style approaches in other domains. I consider boosting such research to be the most interesting outcome of this paper, but would guess there's a bunch of tacit knowledge or more general statements that could be made, beyond what's come out of the specific setting of group composition. For example, I think a lot of the discussion in the rebuttals re whether compact proofs are a reasonable technique was valuable, and would be good to communicate somewhere.

---

### Official Review · Reviewer_QyuD · 2024-11-04

**Soundness:** 3
**Presentation:** 3
**Contribution:** 2
**Rating:** 6
**Confidence:** 3

**Summary:**

This paper contributes to a recent line of work aiming to mechanistically understand the computations performed by neural networks trained on the symmetric group. It takes a step towards this goal by developing an interpretation of the model's computation that can be formally translated into a compact proof of model performance. This compact proof of performance can be measured against the actual performance of the network, serving as a quantitative measure of the quality of a proposed interpretation. The interpretation proposed by the authors is based on their notion of rho-sets, which corresponds to an interpretation of the network learning to become approximately equivariant in each of its inputs. The rho-set interpretation gives rise to a compact proof that can account for the behaviour of approximately half of the models they train. Previous work on the symmetric group came to differing conclusions based on "irrep sparsity" and cosets, but for the approximately 50% of models the rho-set interpretation can account for, it is able to unify the differing interpretations of previous works and show that they are not at odds in these cases. For the other half of models which they are unable to account for with their interpretation, the compact proofs fail to attain non-vacuous bounds. Thus, the authors argue that compact proofs are a concrete way to measure the validity of one's interpretation of neural network computations.

**Strengths:**

1. Compact proofs are a new way of supporting model interpretations and it seems like they could be interesting, since as the paper states, valid compact proofs can be generated from interpretations one is certain of.
2. For the half of the models that the rho-set interpretation works for, explaining how to reconcile the irrep sparsity and cosets interpretation is helpful.

**Weaknesses:**

1. While compact proofs are an interesting way to approach interpretability, it's unclear whether they could be used to help interpret neural network solutions for datasets where no or limited explicit information is known about the distribution it was sampled from (e.g. any language task, CIFAR-10, etc.).
2. The fact that the compact proofs derived in this work only get approximately a 50% success rate is concerning, as it implies that the framework using rho-sets is possibly not general enough.
3. Unifying is too strong a word to use in the title when rho-set compact proofs only work approximately 50% of the time.
4. As someone who has familiarity with representation theory, it's still quite hard to understand the rho-set construction and specifically how it can be identified within the network. I must believe that it works since you can write a compact proof (verifier) that empirically matches the network's performance around half the time. However, since it's unclear how you arrived at this rho-set interpretation by mechanistically inspecting the network, it's not clear how other people can use this to come up with compact proofs for other datasets. If compact proofs are to be useful in the field of interpretability, you should be more clear about how you went about figuring this out. E.g. what are the concrete steps you thought of and experiments you ran to define everything in section 5.1 as well as the observations in Appendix B. Being explicit about these things could greatly help the community understand how to integrate and improve interpretations and contribute to compact proofs.

**Questions:**

1. Can compact proofs be used on datasets without "closed-form" solutions?
2. The way that V_coset is being computed could be fully responsible for the results that cosets have loose vacuous bounds. Reverse engineering the problem qualitatively shows cosets (Stander et al.). I'm wondering if you tried other ways of modeling V_coset? If so, how many other ways did  you try?
3. In section 3.2 why do you take the number of neurons to be equal to the embedding dimension (m)? Is this by chance or necessary for your proofs and interpretation?

---

> ### Author Response · Authors · 2024-11-20
> **Response to reviewer QyuD**
>
> Thank you for your thorough review!
>
>
> > 2. The fact that the compact proofs derived in this work only get approximately a 50% success rate is concerning, as it implies that the framework using rho-sets is possibly not general enough.
> > 3. Unifying is too strong a word to use in the title when rho-set compact proofs only work approximately 50% of the time.
>
> We agree that we do not have a complete understanding of the ~50% of models for which we are unable to obtain nonvacuous bounds. More precisely, for these models, we cannot explain how the individual neurons together contribute to a complete algorithm (e.g. when Observation B.2.4 doesn't hold).
> Nonetheless, most of our observations do hold consistently. When restricting our attention to individual neurons, our observations and explanations via $\rho$-sets are consistently valid and in that case they indeed unify the observations in previous work, as shown in Section 7 and Lemma F.3.
> To put things into perspective, we would like to stress that the baseline explanation presented in previous work is not rigorous enough to yield any nonvacuous bound. Attempts to make it more rigorous have also yielded vacuous bounds as we discussed in the paper (Appendix C). We agree with reviewer ZpTx that this is a strength of the compact proof approach: The fact that for ~50% of models we get a vacuous bound helped us to discover that we don't sufficiently understand these specific models.
>
>
>
>
> > 4. As someone who has familiarity with representation theory, it's still quite hard to understand the rho-set construction and specifically how it can be identified within the network. \[...\] since it's unclear how you arrived at this rho-set interpretation by mechanistically inspecting the network, it's not clear how other people can use this to come up with compact proofs for other datasets.
>
> We added section B.3 to the appendix, detailing the step-by-step process by which we discovered the rho-set circuit. We describe concrete tests that were used to validate each step, so that the reader could rediscover the circuit themselves.
>
> > 1. While compact proofs are an interesting way to approach interpretability, it's unclear whether they could be used to help interpret neural network solutions for datasets where no or limited explicit information is known about the distribution it was sampled from (e.g. any language task, CIFAR-10, etc.).
>
>  > Q1: Can compact proofs be used on datasets without "closed-form" solutions?
>
> In this case you need to specify what you are trying to measure or what the dataset is that you care about. The behaviour/mechanism you try to explain typically has a specific dataset that exhibits this behaviour. So you could restrict your dataset entirely to these specific examples and, if desired, apply the brute force method for all other inputs to attain a compact explanation for the entire dataset.
>
> For example, it may be possible to explain GPT2's Indirect Object Identification (IOI) ([Wang et al. 2022](https://arxiv.org/abs/2211.00593)) circuit within the compact proofs framework by constructing a guarantee that GPT2 outputs the correct indirect object for a large proportion of the samples in a synthetic dataset. (E.g. all sequences of the form "X and Y went to the store. Y gave a store to", where X and Y vary over all tokens corresponding to names of people.) A guarantee of this form would be over a "closed-form" distribution instead of the entire training corpus, yet would still provide a meaningful explanation of how a realistic model performs a specific task that is more precise than existing work. We believe examples such as these are interesting directions for future work.

---

> > ### Author Response · Authors · 2024-11-20
> >
> > >Q2: The way that V_coset is being computed could be fully responsible for the results that cosets have loose vacuous bounds. Reverse engineering the problem qualitatively shows cosets (Stander et al.). I'm wondering if you tried other ways of modeling V_coset? If so, how many other ways did you try?
> >
> > The approach presented in the paper is the only one we tried. In general, we are aware of only one high-level strategy that turns an interpretation into a bound: Bounding the margin of the logits using idealized weights of the model. The interpretation yields the idealized weights. It could be the case that another strategy would yield better bounds---we mention this shortcoming in section 7.2. Ultimately, the coset interpretation of Stander et al. leaves open a few details (for example the bias of the unembedding). If these details were more thoroughly understood, it could be more plausible to get better bounds.
> >
> >
> >
> > >Q3: In section 3.2 why do you take the number of neurons to be equal to the embedding dimension (m)? Is this by chance or necessary for your proofs and interpretation?
> >
> > That the embedding dimension is equal to the pre-activation dimension (i.e., that $\mathbf{W}_l,\mathbf{W}_r\in\mathbb{R}^{m\times m}$ are square) is an arbitrary architectural decision with no significant implications. The proofs and interpretation presented in the paper work just as well when the two dimensions are unequal. That is, we could instead define $\mathbf{W}_l,\mathbf{W}_r\in\mathbb{R}^{m_1\times m_2}$ and $\mathbf{E}_l,\mathbf{E}_r\in\mathbb{R}^{m_2\times |G|}$ where $m_1$ is the pre-activation dimension and $m_2$ is the embedding dimension; the choice we make in the paper that $m_1=m_2=m$ is purely for convenience.
> >
> > After "folding" the linearities $\mathbf{W}_l,\mathbf{W}_r$ into the embeddings $\mathbf{E}_l,\mathbf{E}_r$ (Eq. 1), the $m_2$ dimension is contracted and we are left with only $m_1=m$. Neurons are defined as the coordinates of the pre-activation space, and thus the number of neurons is equal to $m_1=m$ by definition.

---

> > > ### Author Response · Authors · 2024-11-26
> > >
> > > We wanted to follow up to determine whether your concerns are all properly addressed. If you have any remaining questions/comments, please let us know. Thank you again for the careful review of our paper!

---

> > > > ### Comment · Reviewer_QyuD · 2024-11-27
> > > > **Official Comment by Reviewer QyuD**
> > > >
> > > > I would like to thank the authors for their detailed responses and updates to the paper, which improve its clarity and reproducibility. I am excited by the idea of the compact-proof framework and its potential to offer rigorous, quantitative evaluations of neural network interpretations. The addition of Appendix B.3 is especially appreciated, as it begins to address some concerns about reproducibility and methodology. I remain cautiously optimistic about this framework’s broader applicability.
> > > >
> > > > I appreciate the clear exposition on some of the limitations of the work, but I think that the claim of unification should be toned down. I remain unconvinced by the claim of "unifying" prior interpretations, for the following reasons:
> > > > - The rho-sets interpretation accounts for only approximately 50% of the models (trained on S5), leaving a significant proportion unaccounted for. This undermines the claim of unifying prior works.
> > > > - The paper itself acknowledges that rho-sets combine aspects of irrep sparsity and coset concentration and is equivalent to the conjunction. This situates the interpretation at the intersection of prior works, rather than representing a broader "unification" or "union."
> > > >
> > > > Concretely: to better reflect the contributions, I suggest rephrasing the title and framing the paper as a "step towards unifying" prior interpretations. For example:
> > > > - The beginning of the title could be minimally changed to "A Step Towards Unifying and Verifying Mechanistic Interpretations"
> > > > - You could explicitly state in the abstract that the work focuses on models trained on S5 (as also suggested by reviewer ZpTx) and serves as a proof of concept for compact proofs.
> > > >
> > > > I'm willing to increase my score if the authors just slightly tone it down. Again, I think "a step towards" and a slight modification to the abstract would be sufficient to increase my score.
> > > >
> > > > A few questions:
> > > > - The identification of failure modes like  a-bad and  rho-bad is valuable, but as tasks become more complex (e.g., language or vision datasets) and architectures more varied, the number of potential failure modes could grow significantly. Can the current approach adapt to this diversity?
> > > > - Have the authors considered how this framework would transfer to a different architecture? Would it require starting from scratch to develop task-specific or architecture-specific interpretations/compact proofs?

---

> > > > > ### Comment · Reviewer_QyuD · 2024-11-27
> > > > > **Official Comment by Reviewer QyuD**
> > > > >
> > > > > As I understand it, the intended contribution of this paper is to provide a proof of concept for compact proofs, by applying the framework introduced in [1]. I have some suggestions on the general presentation of the paper by framing the paper as a walkthrough or "tutorial", which I think would enhance its value to the interpretability community. Reviewer ZpTx already started giving great suggestions to increase general clarity which I'm glad to see the authors have engaged with. At this point, I'm not sure how much the paper can change for the camera-ready version, so I don't expect the authors to implement changes to the overall text based on my additional suggestions at this point. That said, I think explicitly detailing the thought process and concrete steps involved in reverse-engineering the network and translating it into compact proofs, the paper could become a critical resource for researchers aiming to apply this framework to new tasks. Appendix B.3 is a great addition, and I'd be happy to continue the discussion on how this could be enhanced.
> > > > >
> > > > > I think with this framing, the paper would provide a rigorous proof of concept while setting realistic expectations for the compact-proof approach's current and future capabilities. Overall, I remain optimistic with this paper and would hope that a camera-ready version, should it be accepted, adds more to the appendix to help future researchers contribute to the agenda since I think it will require many minds.
> > > > >
> > > > > [1] J. Gross, "Compact Proofs of Model Performance via Mechanistic Interpretability" (2024)

---

> > > > > > ### Author Response · Authors · 2024-11-29
> > > > > >
> > > > > > Thank you for your followup comments!
> > > > > >
> > > > > > > I suggest rephrasing the title and framing the paper as a "step towards unifying" prior interpretations
> > > > > >
> > > > > > We edit the abstract and introduction to say "a step towards unifying". We change the title of the paper to "Towards a unified and verified understanding of group-operation networks", which we hope better reflects our paper's contribution.
> > > > > >
> > > > > > > You could explicitly state in the abstract that the work focuses on models trained on S5
> > > > > >
> > > > > > We added this to the revised abstract.
> > > > > >
> > > > > > > The identification of failure modes like a-bad and rho-bad is valuable, but as tasks become more complex (e.g., language or vision datasets) and architectures more varied, the number of potential failure modes could grow significantly. Can the current approach adapt to this diversity?
> > > > > >
> > > > > > Whether the number of potential failure modes will grow for more complex tasks and architectures remains to be seen. Our intuition is that the diversity of solutions that we've found in trained models (including $\mathbf{a}$-bad and $\rho$-bad) is in part due to the shallowness of the architecture we use. It seems that a large fraction of these models are converging to suboptimal local minima and that this is responsible for many of the deviations from the ideal $\rho$-set circuit (e.g., see Figure 4 -- models for which $\mathbf{a}$-bad occurs, i.e. $\mathbf{a}$ is nonconstant, are precisely those with inferior cross-entropy loss and higher weight norm). For deeper and more overparameterized models, we expect this diversity in training runs to be less likely. For example, the Git Re-basin paper [1] finds that independently trained ResNet models nearly all converge to the same basin (modulo neuron permutation).
> > > > > >
> > > > > > In any case, in practice, it is often sufficient to interpret only a single trained model instance (the one being deployed), instead of having a family of interpretations that covers all possible training runs. Indeed, much of the interpretability work on more complex models focuses on just a single instance. In this case, the diversity of potential failure modes across training runs isn't as much of a concern -- one need only deal with the single instance of interest.
> > > > > >
> > > > > > [1] Samuel K. Ainsworth, Jonathan Hayase, Siddhartha Srinivasa. "Git Re-Basin: Merging Models modulo Permutation Symmetries". ICLR 2023.
> > > > > >
> > > > > > > Have the authors considered how this framework would transfer to a different architecture? Would it require starting from scratch to develop task-specific or architecture-specific interpretations/compact proofs?
> > > > > >
> > > > > > For now, we indeed start from scratch to construct interpretations and compact proofs for each new task and architecture. However, our hope is that we may find common patterns that allow compact proof techniques to be shared between tasks/architectures, in the same way that mechanistic interpretability work has found circuits shared between a variety of models.
> > > > > > For example, the set-up in [2] uses one layer attention-only transformers trained on the max-of-k task. But many of the techniques can be transferred to more general tasks. For example, constructing bounds using the SVD decomposition of the QK-matrix is described in appendix G of [2] and this could transfer well to other situations where the QK-matrix would be of approximately low rank.
> > > > > >
> > > > > > [2] J. Gross et al. "Compact Proofs of Model Performance via Mechanistic Interpretability." 2024.
> > > > > >
> > > > > >
> > > > > > >As I understand it, the intended contribution of this paper is to provide a proof of concept for compact proofs, by applying the framework introduced in [1]. I have some suggestions on the general presentation of the paper by framing the paper as a walkthrough or "tutorial", which I think would enhance its value to the interpretability community.
> > > > > >
> > > > > > We see this as one of the two main contributions, the other being the more accurate reverse engineering of the group composition algorithm. We cannot modify the paper for the submission anymore, but we might choose to present the material in another form (e.g. a blog post), in which case we could incorporate new suggestions. We would be appreciative of your thoughts here.

---

> > > > > > > ### Author Response · Authors · 2024-12-02
> > > > > > >
> > > > > > > Since today is the last day for which reviewers can respond, we just wanted to follow up one more time about whether our rephrasing of the title, abstract, and introduction properly addresses your concern about the "unifying" wording being too strong. Thank you!

---

> > > > > > > > ### Comment · Reviewer_QyuD · 2024-12-02
> > > > > > > > **Official Comment by Reviewer QyuD**
> > > > > > > >
> > > > > > > > Thanks for your thoughtful responses and the updates to the paper to increase clarity and reflect contributions! I'm cautiously optimistic about the compact proofs approach, but still a bit concerned about scalability and transferability.
> > > > > > > >
> > > > > > > > I agree with reviewer ZpTx that boosting such research to be the most interesting outcome of the paper.
> > > > > > > > > For the camera ready, it might be nice to add a more informal appendix giving advice to researchers trying to use compact proof-style approaches in other domains. I consider boosting such research to be the most interesting outcome of this paper, but would guess there's a bunch of tacit knowledge or more general statements that could be made, beyond what's come out of the specific setting of group composition. For example, I think a lot of the discussion in the rebuttals re whether compact proofs are a reasonable technique was valuable, and would be good to communicate somewhere.
> > > > > > > >
> > > > > > > > In particular, I think disseminating such information will allow the approach to be validated more quickly.
> > > > > > > >
> > > > > > > > I've updated my score.

---

### Author Response · Authors · 2024-11-20
**Changes present in revised version**

We would like to thank all reviewers for the insightful reviews and comments.

First of all, we record the following changes in the revised submission:
- We have modified the last paragraph of the introduction to better explain the notion of compact proof
- We edited Section 4.1, Section 5.1, Section 7.1/7.2 for clarity and to motivate our results better
- We added Appendix B.3 walking through our process for arriving at the $\rho$-sets interpretation.
- We added Appendix C which contains experiments that use conventional methods to confirm our observations/interpretation
- We added Appendix J which discusses bounds on cross-entropy loss instead of accuracy
- We added experiments for $A_5$ in Appendix K.1
- Minor fixes to address reviewer comments

---

> ### Author Response · Authors · 2024-11-20
>
> Furthermore, we would like to make a few general comments about compact proofs.
>
> The case of larger realistic models such as LLMs are the cases we ultimately care about -- we think of our work as proof of concept. We believe that exploring to what extent compact proofs can be scaled to more realistic settings is an important direction for future research. Less strict variants of compact proofs, e.g. using sampling or restricting to certain subsets of interest, could be a compromise that is easier to scale.
>
> If we think of interpretations sitting on a spectrum between "worst-case" and "average-case" behaviour over the unexplained portion of model weights, most mechanistic interpretability research falls into the latter category, whereas our work covers the former. We think it is valuable to explore the other end of the spectrum and see how far one could take it.
>
> We find that existing methods to evaluate the faithfulness or compactness of an interpretation in a rigorous and quantative way are limited and have shortcomings (see the references mentioned in the introduction). In fact, in the literature, the notion of interpretation itself is still rather vague. We believe these notions are important to make precise to establish a more rigorous science of interpretability. The compact proofs approach addresses these points, but it is not yet clear how well it will scale. This leaves the question open of how we are to measure these quantities and how we should formalize the notion of an interpretation.

---

### Author Response · Authors · 2024-11-27
**New revision**

Thanks again to all reviewers for their time and effort.
We've submitted another revision addressing the points made in the most recent round of comments:

- We softened the claim of "unification" in the title, abstract, and intro in response to Reviewer QyuD
- The abstract now clarifies that the main text focuses on only the group $S_5$.
- The abstract also notes the non-asymptotic 3x speedup for $S_5$ bounds
- We cut the old Fig 1 and substitute a new one illustrating example $\rho$-sets
- We added experiment results for the group $S_4$ (Figure 10)
- We've implemented the writing changes suggested by ZpTx for increased clarity. In particular:
  - We moved the content of the old Section 4 after the old Section 5
  - We added more explanation of compact proofs to the introduction

---

### Meta-Review · Area_Chair_wGCZ · 2024-12-19

**Metareview:**

The paper provides a novel mechanistic interpretation of how a single-layer fully-connected network performs group composition in $S_5$. This explanation extends and unifies the explanations proposed in prior work, which the authors argue do not account for parts of model behavior. The authors then convert their mechanistic explanation to a compact proof of model performance, i.e. a computable bound on model accuracy. The authors show that their approach results in a non-vacuous bound 50% of the time, and can be computed 3 times faster than a brute-force bound which evaluates the model accuracy on all possible inputs.

Strengths:
-  The authors provide a novel interesting mechanistic interpretation in the setting that was considered by prior work (learning group composition)
- The proposed explanation unifies previously proposed explanations
- The authors convert their explanation to a compact proof of performance
  + The proof of performance is better for the proposed explanation compared to prior work
- The paper is generally well-written, the figures are of high quality

Weaknesses:
- The paper is dense and multiple reviewers mentioned concerns with clarity; some of the concerns have been addressed in the rebuttal phase
- The paper only considers a single group $S_5$
- The proposed explanation only results in a non-vacuous performance proof 50% of the time
  + The proposed explanation is not always correct (is not correct for some of the models)
- It is not clear how the methodology of the paper can be generalized to realistic models beyond toy settings

Decision recommendation: I believe this is a high quality paper and I recommend to accept it. Despite the limitations, the paper makes a strong contribution to the mechanistic interpretability literature.

**Additional Comments On Reviewer Discussion:**

All three reviewers are recommending to accept the paper with scores 6, 8, 8. The authors provided a detailed rebuttal, and the reviewers engaged in a discussion with the authors. As a result, two of the reviewers increased their scores, and the authors made significant updates to the paper to improve clarity and also to make the wording more precise.

---

### Decision · Program_Chairs · 2025-01-22

Accept (Spotlight)